# Mechanistic insight into the competition between interfacial and bulk reactions in microdroplets through $N_2O_5$ ammonolysis and hydrolysis

Ye-Guang Fang[1,2], Bo Tang[1], Chang Yuan[1], Zhengyi Wan[3], Lei Zhao[1], Shuang Zhu[1], Joseph S. Francisco [3] ✉, Chongqin Zhu [1] ✉ & Wei-Hai Fang [1]

Reactive uptake of dinitrogen pentaoxide ($N_2O_5$) into aqueous aerosols is a major loss channel for $NO_x$ in the troposphere; however, a quantitative understanding of the uptake mechanism is lacking. Herein, a computational chemistry strategy is developed employing high-level quantum chemical methods; the method offers detailed molecular insight into the hydrolysis and ammonolysis mechanisms of $N_2O_5$ in microdroplets. Specifically, our calculations estimate the bulk and interfacial hydrolysis rates to be $(2.3 \pm 1.6) \times 10^{-3}$ and $(6.3 \pm 4.2) \times 10^{-7}$ $ns^{-1}$, respectively, and ammonolysis competes with hydrolysis at $NH_3$ concentrations above $1.9 \times 10^{-4}$ mol $L^{-1}$. The slow interfacial hydrolysis rate suggests that interfacial processes have negligible effect on the hydrolysis of $N_2O_5$ in liquid water. In contrast, $N_2O_5$ ammonolysis in liquid water is dominated by interfacial processes due to the high interfacial ammonolysis rate. Our findings and strategy are applicable to high-chemical complexity microdroplets.

Dinitrogen pentaoxide ($N_2O_5$) has long been recognized as an important reactive intermediate in the atmospheric chemistry of nitrogen oxide and nitrate aerosols, and it plays a key role in night-time atmospheric chemistry[1]. The atmospheric significance of $N_2O_5$ stems from its role as a temporary reservoir for $NO_x$ ($NO_x = NO + NO_2$) species, significantly impacting the levels of atmospheric ozone ($O_3$), hydroxyl radicals (·OH) and methane ($CH_4$)[2,3]. Several studies using air quality modelling have shown that tropospheric $N_2O_5$ affects oxidant levels on urban, regional and global scales[4,5].

Over the past few decades, the reactive uptake of $N_2O_5$ on aerosols has been widely considered one of the most influential processes in heterogeneous atmospheric chemistry[1,6,7]. Between 25 and 41% of the $N_2O_5$ in the troposphere is thought to be removed via reactive uptake by aerosols[8,9]. Due to its importance, the reactive uptake of $N_2O_5$ by aqueous aerosols has been extensively studied theoretically[10–13] and experimentally[14,15]. To date, it is unclear whether $N_2O_5$ hydrolysis occurs near the aerosol surface or throughout the aerosol volume due to the lack of molecular understanding of the hydrolysis process. Recently, Galib et al. performed molecular dynamics (MD) simulations of the reactive uptake of $N_2O_5$ by liquid water using a neural network-based reactive model, and they concluded that interfacial processes, not bulk phase processes, determine the observed uptake coefficient[12]. The results from experiments[15] and calculations[13] have questioned this conclusion. Notably, the model used in the simulations investigated by Galib et al. was constructed from ab initio molecular dynamics (AIMD) simulations with a low-level quantum chemical method (revPBE-D3/ MOLOPT-DZVP). High-level quantum chemical methods for MD

[1]Key Laboratory of Theoretical and Computational Photochemistry, Ministry of Education, College of Chemistry, Beijing Normal University, Beijing, P. R. China. [2]Laboratory of Theoretical and Computational Nanoscience, CAS Key Laboratory of Nanosystem and Hierarchical Fabrication, CAS Centre for Excellence in Nanoscience, National Centre for Nanoscience and Technology, Beijing, P. R. China. [3]Department of Chemistry, University of Pennsylvania, Philadelphia, PA, USA. ✉e-mail: frjoseph@sas.upenn.edu; cqzhu@bnu.edu.cn

simulations are needed to gain increasingly reliable insights into chemistry.

Conversely, recent studies have suggested that atmospheric ammonia can catalyse reactions and promote the transformation of chemical species in the atmosphere[16,17]. In addition, ammonolysis and hydrolysis processes are essential for the removal of important atmospheric species[18,19]. Furthermore, measurements of $NH_3$ concentrations in the troposphere have revealed unexpectedly high amounts[20], and $NH_3$ has been detected several times with a maximum mixing ratio of ~30 pptv[20] and up to 1.4 ppbv in popular locations[21], indicating that ammonolysis may play an important role in the elimination of $N_2O_5$ from the atmosphere[19,22]. Unfortunately, the effect of $NH_3$ on the reactive uptake of $N_2O_5$ by atmospheric aerosols remains unclear, although recent calculations have implied that the catalytic roles of $NH_3$ and $H_2O$ are negligible in determining the atmospheric fate of $N_2O_5$ via gas phase hydrolysis and ammonolysis[23,24].

To simulate the hydrolysis and ammonolysis of $N_2O_5$ in liquid water using high-level quantum chemical methods, we develop a strategy based on stepwise multisubphase metadynamics (SMS-MetaD). Specifically, we employ two-step MD simulations. In the first step, numerous independent (MetaD-biased) quantum mechanics/molecular mechanics (QM/MM) MD simulations with large QM regions are performed using low-level quantum chemical methods to determine reaction pathways. In the second step, high-level (MetaD-biased) QM/MM MD simulations with small QM regions are then conducted to acquire accurate free energy profiles and reaction rates. The QM method is used to depict the molecules involved in chemical reactions to reduce the computational cost, enabling quantitative studies of the thermodynamics and kinetics of $N_2O_5$ uptake. Our QM/MM MD simulations at the PBE0-D3/MOLOPT-DZVP-SR level combined with enhanced sampling techniques show that the predicted bulk hydrolysis rate is consistent with experiments and is four orders of magnitude faster than the interfacial hydrolysis rate. Additionally, the rate of ammonolysis of $N_2O_5$ in liquid water is five orders of magnitude faster than that of bulk hydrolysis. The results reveal a complete quantitative picture of the reactive uptake of $N_2O_5$ by atmospheric aerosols with or without $NH_3$.

## Results and discussion
### Gas-phase reaction

As a preliminary step, we explored the production of $HNO_3$ via $N_2O_5$ hydrolysis and ammonolysis using quantum chemical calculations. The catalytic effects of $H_2O$ and $NH_3$ on these reactions were considered. The energy barriers ($\Delta E_b$) for these reactions calculated at various levels of theory, including the benchmark CCSD(T)/aug-cc-pVTZ//PBE0/6-31 + G** level, were compared, as illustrated in Fig. 1. For the $N_2O_5 + H_2O$ reaction and the $N_2O_5 + 2H_2O$ reaction, two and one reaction pathways for $N_2O_5$ hydrolysis were characterized, respectively, as shown in Fig. 1aI–III. Moreover, the hydrolysis of $N_2O_5$ with the presence of $NH_3$ was investigated (Fig. 1aIV). These computed reactions were consistent with those previously reported[23]. All the hybrid functionals (including B3LYP, PBE0, M06, B2PLYP) evaluated showed similar performance characteristics, and they outperformed the generalized gradient approximation (GGA) functionals (BP86, BLYP, PBE, revPBE). Specifically, the $\Delta E_b$ calculated using hybrid functionals ranged from 13.7 to 15.6 kcal mol$^{-1}$, 23.6 to 26.8 kcal mol$^{-1}$, 13.9 to 18.0 kcal mol$^{-1}$ and 14.8 to 20.7 kcal mol$^{-1}$ for pathways 1W-1, 1W-2, 2W and W-A, respectively (blue bars in Fig. 1b, d). GGA functionals (red bars in Fig. 1b, d) typically exhibited lower $\Delta E_b$ values than hybrid functionals. Of these GGA functionals, revPBE performed the best; it had energy barriers of 8.7, 19.6, 13.9, and 9.8 kcal mol$^{-1}$ for pathways 1W-1, 1W-2, 2W and W-A, respectively. Using the CCSD(T) method as a standard, these values were accordingly 12.0, 10.5, 8.4 and 12.8 kcal mol$^{-1}$ lower than those calculated at the CCSD(T)/aug-cc-pVTZ//PBE0/6-31 + G** level.

With the hydrolysis of $N_2O_5$, the ammonolysis of $N_2O_5$ with or without the presence of $H_2O$ in the gas phase was studied. Two different reaction pathways for the $N_2O_5 + NH_3$ reaction were characterized, as shown in Figs. 1cV, cVI; we identified a reaction pathway for the monohydrate system, as shown in Fig. 1cVII. Pathway 1A-1 and pathway A-W were equivalent to those reported by Sarkar and Bandyopadhyay[24]. Figure 1d demonstrates that all the hybrid functionals that we evaluated performed much better than the GGA functionals. Notably, the explored GGA functionals performed poorly when calculating the energy barrier for the ammonolysis of $N_2O_5$. These GGA functionals yielded almost zero $\Delta E_b$ values for pathways 1A-1 and A-W; however, the hybrid functionals yielded noticeably increased energy barriers. Further gas-phase calculations show that the D3 correction has little effect on the reaction barriers (Supplementary Fig. S4), which justifies the use of the vdW corrections in condensed-phase calculations.

### Solvation and hydrolysis

The solvation and adsorption of $N_2O_5$ in water was extensively studied previously. For instance, recent MD simulations using a data-driven many-body model of coupled-cluster accuracy showed that the equilibrium density profile of $N_2O_5$ was inhomogeneous near the air–water interface[13]. Herein, by using the umbrella sampling technique associated with classical MD simulations, we calculated the free energy profile for the transfer of a $N_2O_5$ molecule from the gas phase across the air–water interface into bulk water. As shown in Supplementary Fig. S5, as $N_2O_5$ moved from the gas phase towards the bulk phase, the free energy first decreased from 0 to −2.3 kcal mol$^{-1}$ and subsequently increased to a plateau of −1.5 kcal mol$^{-1}$. The minimum at $z = 0$ reflected the preferential location of $N_2O_5$ at the air–water interface, which was in agreement with previous studies[12,13,25,26]. For the definition of the air–water interface, the widely accepted 10–90 thickness is used[27–30].

Exploring sufficiently large inhomogeneous systems using standard conventional AIMD simulations is a challenge, and the problem is more severe for high-level quantum chemical methods. To overcome these limitations, we performed QM/MM MD simulations using the stepwise multisubphase space metadynamics (SMS-MetaD) method[31] to simulate the hydrolysis of $N_2O_5$ at the air–water interface and inside the bulk. To determine the reaction pathways, numerous independent MetaD-biased QM/MM MD simulations with large QM regions were performed at the PBE-D3/DZVP-MOLOPT-SR level of theory in step 1. In these simulations, initial structures were selected randomly, and coarse Gaussian potentials were deposited. The initial structures in step 2 were created by selecting numerous configurations from the simulated trajectories in step 1. To acquire accurate free energy profiles, high-level QM/MM MD simulations with a small QM region were conducted in step 2. The QM method at the PBE0-D3/DZVP-MOLOPT-SR level of theory was used to depict the molecules participating in chemical reactions that contained the $N_2O_5$ molecule (see "Methods" for details).

Three conversion mechanisms for the hydrolysis of $N_2O_5$ at the air–water interface and within bulk water were identified by MetaD-biased QM/MM MD simulations. (i) Upon the splitting of a $H_2O$ molecule, the hydroxyl (OH) group combined with the $NO_2$ motif, and the hydrogen (H) group transferred to the $NO_3$ motif via a characteristic loop-structure, forming two $HNO_3$ molecules (molecular mechanism) (Fig. 2a and Supplementary Movies S1 and S2). (ii) A water molecule split into two groups—an OH and an H—which bound to the $NO_2$ motif of the $O_2NONO_2$ and another $H_2O$ molecule, respectively, forming $H_3O^+$, $NO_3^-$ and $HNO_3$ (ionic mechanism) (Fig. 2b and Supplementary Movie S3). (iii) A water molecule reacted with the $O_2NONO_2$ to form an intermediate $H_2ONO_2^+$ and $NO_3^-$, and then $H_2ONO_2^+$ proceeded to react with $H_2O$ to generate $HNO_3$ and $H_3O^+$ (stepwise ionic mechanism) (Fig. 2c and Supplementary Movie S4). Notably, the stepwise ionic mechanism was not addressed in Galib's study[12].

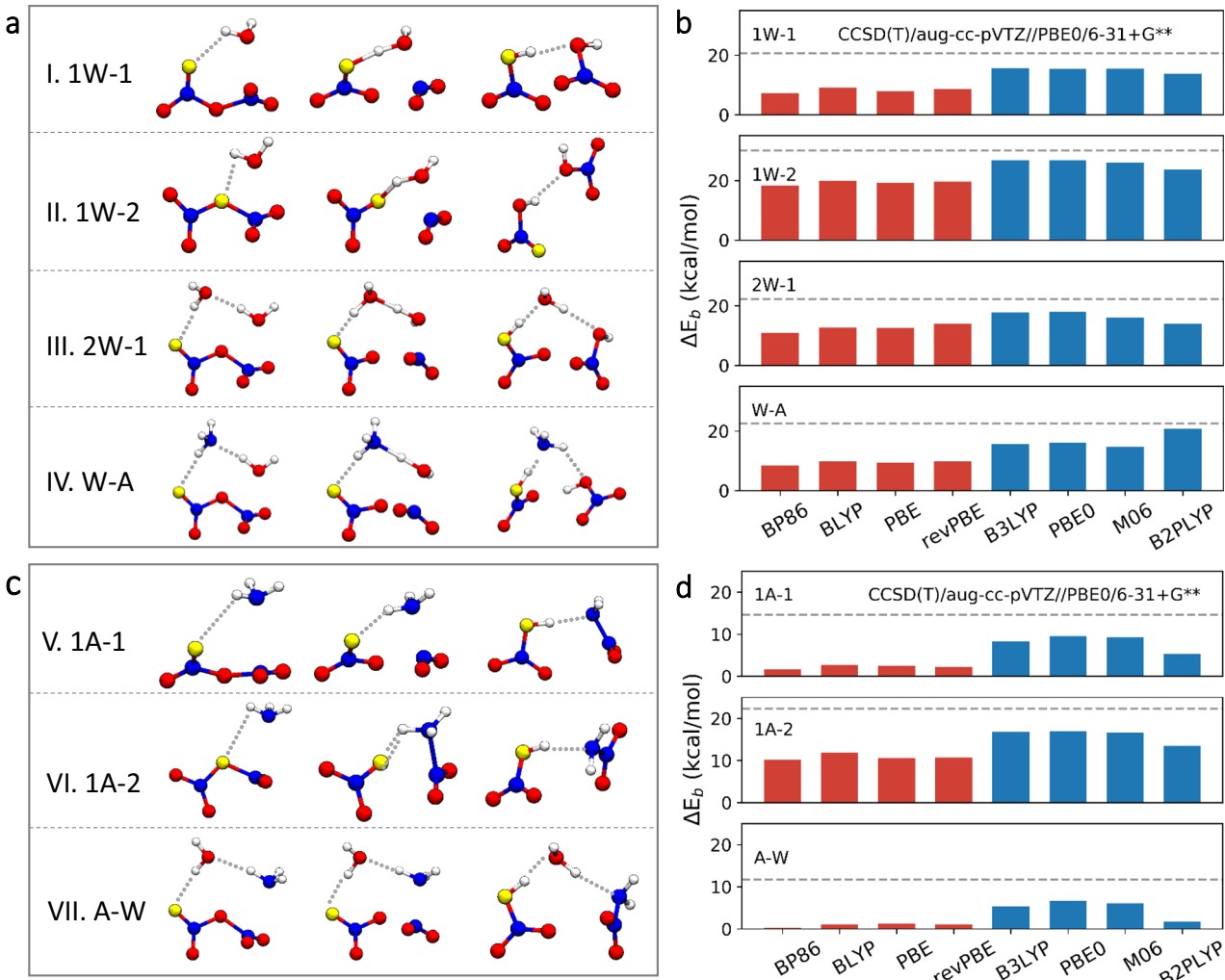

**Fig. 1 | Performance characteristics of various exchange-correlation functionals when calculating the hydrolysis and ammonolysis of N$_2$O$_5$ in the gas phase. a** Structures of the stationary points of the hydrolysis reaction of N$_2$O$_5$ with one water molecule (I and II), two water molecules (III), and one water and one ammonia molecule (IV). **b** Calculated energy barriers $\Delta E_b$ for the hydrolysis reactions of N$_2$O$_5$ for all the functionals considered. **c** Structures of the stationary points of the ammonolysis reaction of N$_2$O$_5$ with one ammonia molecule (V and VI) and with one water and one ammonia molecule (VII). **d** Calculated $\Delta E_b$ for the ammonolysis reaction of N$_2$O$_5$ for all the functionals considered. **a, c** The white and blue spheres represent H and N atoms, respectively. Red and gold spheres represent O atoms. **b, d** The dashed black horizontal line represents the benchmark at the CCSD(T)/aug-cc-pVTZ//PBE0/6-31 + G** level. The red and blue bars in (**b, d**) indicate GGA and hybrid functionals, respectively.

The free energy profiles for the reaction of N$_2$O$_5$ with water monomer at the air−water interface and inside bulk water via the molecular mechanism are displayed in Fig. 3. The free energy barriers for the same reaction pathway were almost the same at the air−water interface and in bulk water, due to the weak polarity of N$_2$O$_5$. An H group of H$_2$O could attach to the terminal or central oxygen atom of O$_2$NONO$_2$ (Supplementary Movies S1 and S2). According to the calculated free energy barriers for N$_2$O$_5$ at the air−water interface and in bulk water, which were -13.4 kcal mol$^{-1}$ and -21.4 kcal mol$^{-1}$, respectively, the former was responsible for the hydrolysis reaction of N$_2$O$_5$ via the molecular mechanism. In addition, various numbers of water molecules were involved in the reaction at the air−water interface via the molecular mechanism (Supplementary Fig. S7 and Supplementary Movies S5 and S6). Surprisingly, unlike the gas phase reactions, the energy barrier for the reaction of N$_2$O$_5$ with water dimer or water trimer at the air−water interface is higher than that for the reaction of N$_2$O$_5$ with water monomer, which is -15.9 kcal mol$^{-1}$ or -19.3 kcal mol$^{-1}$, respectively.

Figure 4 presents free energy profiles for the hydrolysis of N$_2$O$_5$ via ionic and stepwise ionic mechanisms. The overall free-energy differences between the reactants and products indicated that the reaction was thermodynamically favourable. For the ionic mechanism, the reaction pathway involved free energy barriers of -14.9 and -8.1 kcal mol$^{-1}$ at the air−water interface and inside the bulk water, respectively. In contrast, reaction via the stepwise ionic mechanism involved free energy barriers of -13.0 and -9.7 kcal mol$^{-1}$ at the air−water interface and inside the bulk, respectively. In Fig. 4b, a shallow minimum could be found at $d_{CV} = 0.4$ Å, corresponding to the formation of an intermediate of H$_2$ONO$_2^+$. Previous studies[25] have shown that the N atom in the NO$_2$ fragment of N$_2$O$_5$ at the air−water interface and in bulk water is more positively charged than N$_2$O$_5$ in the gas phase. Since the O atom in water molecule is an electron-rich atom, the interaction between the N atoms of N$_2$O$_5$ and O atoms of water molecules leads to the formation of the intermediate H$_2$ONO$_2^+$ ion. Indeed, the intermediate H$_2$ONO$_2^+$ survives for -0.78 and -0.31 ps at the air−water interface and in the bulk water, respectively, according to QM/MM MD simulations of the preexisting intermediate at the PBE0-D3/MOLOPT-DZVP-SR level (Supplementary Fig. S8). In contrast, the average proton transfer time is less than 0.1 ps[32].

Unlike the energy profiles for the N$_2$O$_5$ + $n$H$_2$O reaction with $n = 1$−2 in the gas phase, the hydrolysis reaction of N$_2$O$_5$ could be catalysed at the air−water interface and inside the bulk water. The

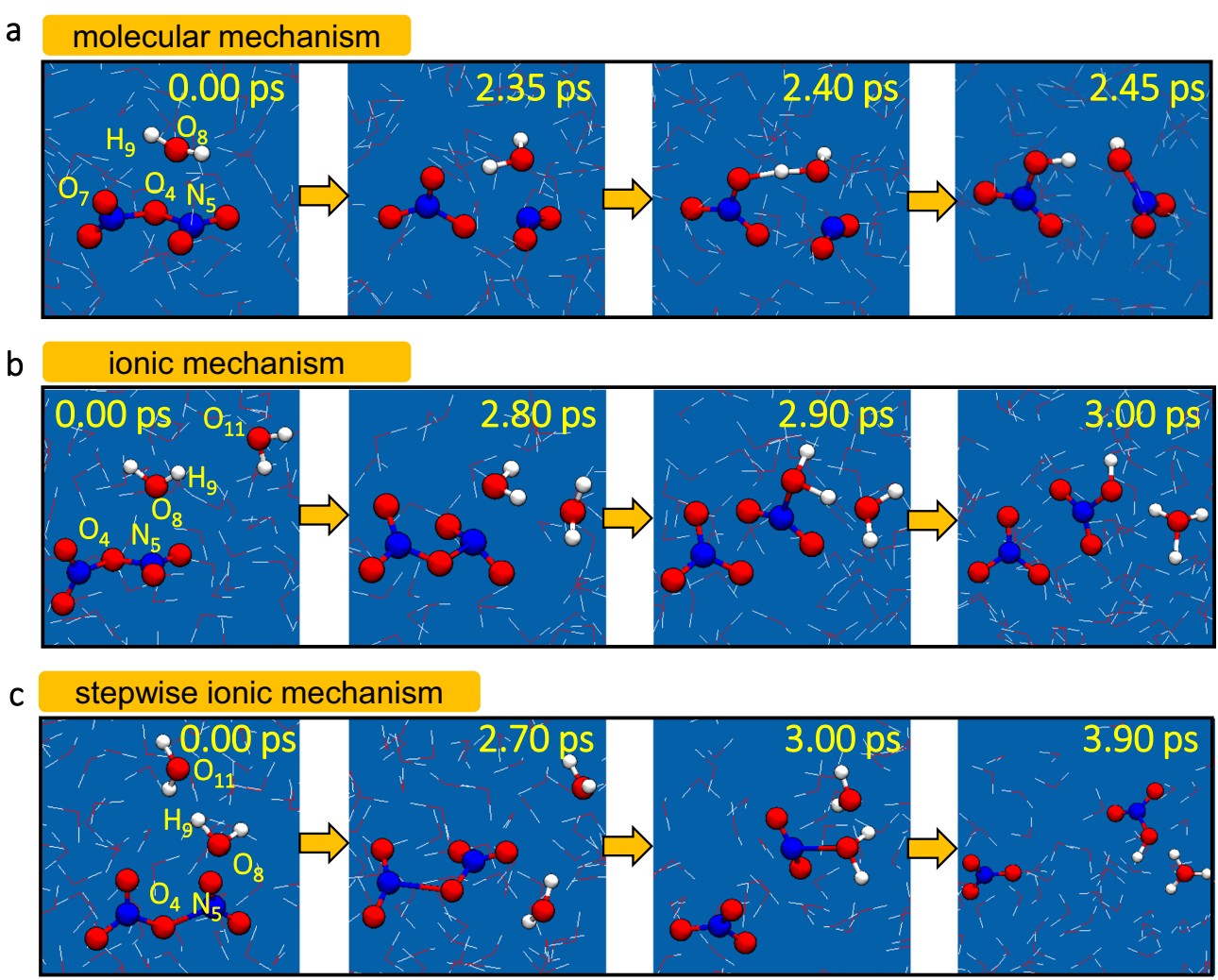

**Fig. 2 | Mechanisms of hydrolysis of $N_2O_5$ in liquid water.** Snapshot structures for $N_2O_5$ hydrolysis in liquid water via the molecular (**a**), ionic (**b**), or stepwise ionic mechanism (**c**) in MetaD-biased QM/MM MD simulations. Corresponding time evolution traits of key bond distances for (**a–c**) are shown in Supplementary Fig S6, respectively.

orders of the calculated free energy barrier for the hydrolysis of $N_2O_5$ via different mechanisms at the air–water interface and in bulk water were ionic mechanism > molecular mechanism ≈ stepwise ionic mechanism and molecular mechanism > stepwise ionic mechanism > ionic mechanism, respectively. These calculations suggested that stepwise ionic and molecular mechanisms played major roles in the hydrolysis reaction of $N_2O_5$ at the air–water interface, whereas the ionic mechanism governed the $N_2O_5$ hydrolysis reaction in bulk water. In addition, we have analysed transition states for $N_2O_5$ hydrolysis via ionic mechanism. Specifically, we investigated 30 configurations belonging to the constrained ensemble with RC ∈ [−0.025, 0.025]. The committors of configurations are narrowly distributed around 0.5 (Supplementary Fig. S9), indicating that the transition state criterion used is good.

**Interfacial and bulk hydrolysis rates**
We estimated the interfacial and bulk hydrolysis rates using the Bennett–Chandler method[33]. Specifically, the hydrolysis rates, $k_h$, were given by the following equation:

$$k_h = k(t)k_h^{TST} \tag{1}$$

where $k(t)$ is the transmission coefficient and $k_h^{TST}$ is the pseudo first-order hydrolysis rate calculated using transition-state theory (TST);

this value was $7.8 \times 10^{-3}$ and $2.1 \times 10^{-6}$ $ns^{-1}$ in the bulk water and at the air–water interface, respectively. Note that TST assumes that the trajectory moves through the transition state undeterred (i.e., the activated trajectory do not recross the transition state). In fact, the active trajectories can recross the transitions state, and $k(t)$ is the fraction of successful trajectories. For those trajectories that are in the transition state at $t = 0$, typical transient dynamics away from it and towards a stable situation will occur in a relatively rapid time $t \sim \tau_{mol}$. From this viewpoint, we can derive the reactive flux correlation function as defined below:

$$k(t) = \langle v(0)\delta[q(0) - q^*]H_B[q(t)]\rangle \tag{2}$$

where, $q(t)$ is the reaction coordinate at time $t$; $v(t)$ is the velocity of that coordinate; $H_B[q(t)]$ is the characteristic function for stable state B, i.e., it is 1 for q(t) > q* and it is zero otherwise. The angle brackets indicate the equilibrium ensemble average over the initial conditions of all degrees of freedom. Based on previous studies, $k(t)$ was estimated to be $0.3 \pm 0.2$[12,13], resulting in bulk and interfacial hydrolysis rates of $2.3 \pm 1.6 \times 10^{-3}$ and $6.3 \pm 4.2 \times 10^{-7}$ $ns^{-1}$, respectively. These rates were in good agreement with those typically inferred in experiments[15,34,35] (ranging from 0.5 to $1.3 \times 10^{-3}$ $ns^{-1}$) and notably slower than those calculated from the neural network model (0.2 $ns^{-1}$)[12]. The discrepancy with respect to the neural network model could be attributed to the

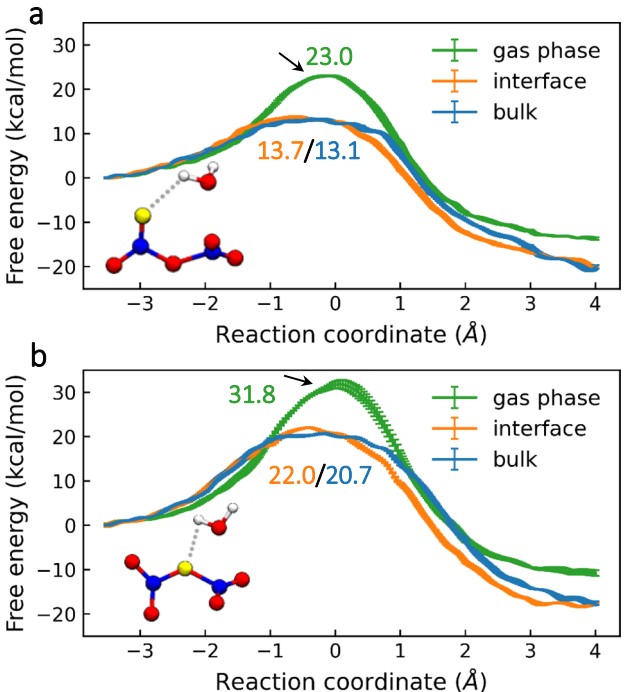

**Fig. 3 | Free energy profiles for the reaction of $N_2O_5$ with water monomer in liquid water via molecular mechanisms.** Free energy profiles for the reaction of $N_2O_5$ with water monomer via the molecular mechanism in the gas phase (green line), at the air–water interface (orange line), or in the bulk water (blue line). During the reaction, the H group of $H_2O$ can attach to the terminal (**a**) or central oxygen atom (**b**) of $O_2NONO_2$. Energy barriers are presented in kcal mol$^{-1}$. The standard deviation of the free energy of the final 8 ps of the MetaD biased QM/MM simulations is used to determine the error bars.

limitations of the low-level quantum chemical method that was used to train the data.

## Ammonolysis

We investigated the ammonolysis of $N_2O_5$ at the air–water interface following the adsorption of $NH_3$ and $N_2O_5$. As the explored low-level quantum chemical methods all performed poorly when calculating the energy barrier for the ammonolysis of $N_2O_5$ in the gas phase (Fig. 1b), high-level quantum chemical methods were required for MD simulations. A similar method to that applied to simulate the hydrolysis of $N_2O_5$ was used, and two-step unbiased QM/MM MD simulations were performed.

We performed ten independent QM/MM MD simulations. For all MD runs, the reactions of $N_2O_5$ with $NH_3$ were directly observed. The simulated evolution of $N_2O_5$ ammonolysis was demonstrated in Fig. 5a–d, which clearly revealed two different mechanisms. (i) Upon the splitting of a $NH_3$ molecule, the $NH_2$ group combined with the $NO_2$ motif and the H group combined with the $NO_3$ motif via a characteristic loop structure, forming $HNO_3$ and $NH_2NO_2$ molecules (molecular mechanism) (Fig. 5a and Supplementary Movie S7). (ii) A $NH_3$ molecule reacted with the $O_2NONO_2$ to form an intermediate $NH_3NO_2^+$ and $NO_3^-$, and the proton was transferred from $NH_3NO_2^+$ to $H_2O$ to generate $H_3O^+$ and $NH_2NO_2$ (stepwise ionic mechanism) (Fig. 5b and Supplementary Movie S8). The time evolution traits of the key bond distances of all ten reactions are shown in Supplementary Figs. S10 and S11.

Figure 5c shows the fraction of unreacted $N_2O_5$ versus simulation time. All reactions occurred within 16.0 ps, and the time scale for the molecular mechanism was slightly shorter than that of the ionic mechanism (Fig. 5d). The average reaction time was ~9.0 ps, which was ~9 orders of magnitude shorter than that estimated in the gas phase

(Supplementary Fig. S12). Among these ten simulations, five reactions followed the molecular mechanism. In four reactions, the H group of $NH_3$ was attached to the central oxygen atom of $O_2NONO_2$, while in the remaining reaction, the H group of $NH_3$ was attached to the terminal oxygen atom of $O_2NONO_2$. Hence, in contrast to $N_2O_5$ hydrolysis, the central oxygen atom of $O_2NONO_2$ was involved in most of the reactions of $N_2O_5$ with $NH_3$ that occurred via the molecular mechanism. Of the five reactions occurring via the molecular mechanism in the ten independent QM/MM MD simulations, four did not involve water molecules, and one involved a single water molecule.

In addition to the molecular mechanism, five out of the ten reactions occurred via the stepwise ionic mechanism. The positively charged $NH_3NO_2^+$ intermediate persisted at a time scale much longer than the conversion process (~0.05 ps). Specifically, in the QM/MM MD simulations at the PBE0-D3/DZVP-MOLOPT-SR level, the lifetime of the intermediate varied between 0.4 ps and 12.0 ps, with an average lifetime of 3.8 ps. Furthermore, structural optimization indicated that the $NH_3NO_2^+$ intermediate was stable (Supplementary Fig. S13). We note that the ammonolysis of $N_2O_5$ mainly occurs mainly at the air–water interface, whereas $N_2O_5$ ammonolysis in bulk water contributes less to the reactive uptake due to the interfacial affinity of $N_2O_5$ and $NH_3$ (Supplementary Fig. S5) as well as the fast reaction rate and short lifetime of $N_2O_5$ in the presence of $NH_3$.

The ultrafast ammonolysis of $N_2O_5$ at the air–water interface indicates that the interfaical ammonoylsis of $N_2O_5$ is barrierless. Under such conditions, the rate constants can be evaluated using collision frequency model. Assuming concentration of the ammonia is $n$ mol L$^{-1}$, the ammonia concentration at the air–water interface is given by

$$c = n \, \exp(\beta\Delta F_b) \tag{3}$$

where $\beta\Delta F_b = 1.54$ is the barrier of $NH_3$ to move from the bulk liquid to the interface[26]. Then, an expression for the collision frequency of each $N_2O_5$ molecule is obtained:

$$k = 1.54 n N_A \pi (r_{NH_3} + r_{N_2O_5})^2 \sqrt{v_{N_2O_5}^2 + v_{NH_3}^2} \tag{4}$$

where $N_A$ is the Avogadro constant, $r_{NH_3} = 0.20 nm$ is the radius of $NH_3$, $r_{N_2O_5} = 0.35 \, nm$ is the radius of $N_2O_5$, $v_{N_2O_5} = 15.8$ nm ns$^{-1}$ is the average velocity of $N_2O_5$, and $v_{NH_3} = 15.9$ nm·ns$^{-1}$ is the average velocity of $NH_3$. Then $k = 2.02 \times 10^{10}$ ns$^{-1}$. In order to have a clear picture of ammonolysis rates as a function of $NH_3$ concentration ($n$), we plotted $k$ against $n$, as shown in Supplementary Fig. S14. The plot clearly shows that the ammonolysis rate increases monotonically with an increase in $n$. Moreover, it is evident that ammonolysis competes with hydrolysis at $NH_3$ concentrations above $1.9 \times 10^{-4}$ mol L$^{-1}$. Due to the low ammonia concentration, ammonolysis does not usually play a role in the $N_2O_5$ decomposition. Note that recent satellite measurements and integrated cross-scale modeling have shown that ammonia tends to accumulate on the surface of cloud droplets[26]. Sometimes high ammonia concentrations may encountered near intense pollution sources, in which case the interfacial ammonolysis of $N_2O_5$ may be important. On the other hand, ammonoylsis of $N_2O_5$ generates nitramide, which in turn may generates $N_2O$ via photochemical processes[36,37] or water catalyzed processes[38]. Thus, interfacial ammonolysis of $N_2O_5$ may be a missing source of the greenhouse gas $N_2O$.

## Implications for $N_2O_5$ reactive uptake

The resistor model[39,40] provided a concise formulation for estimating the reactive uptake coefficient, $\gamma$, of the trace gas. In the model, we assumed that gas-phase diffusion limitations were negligible for the measured size ranges and values of $\gamma$. This framework simplified $\gamma$

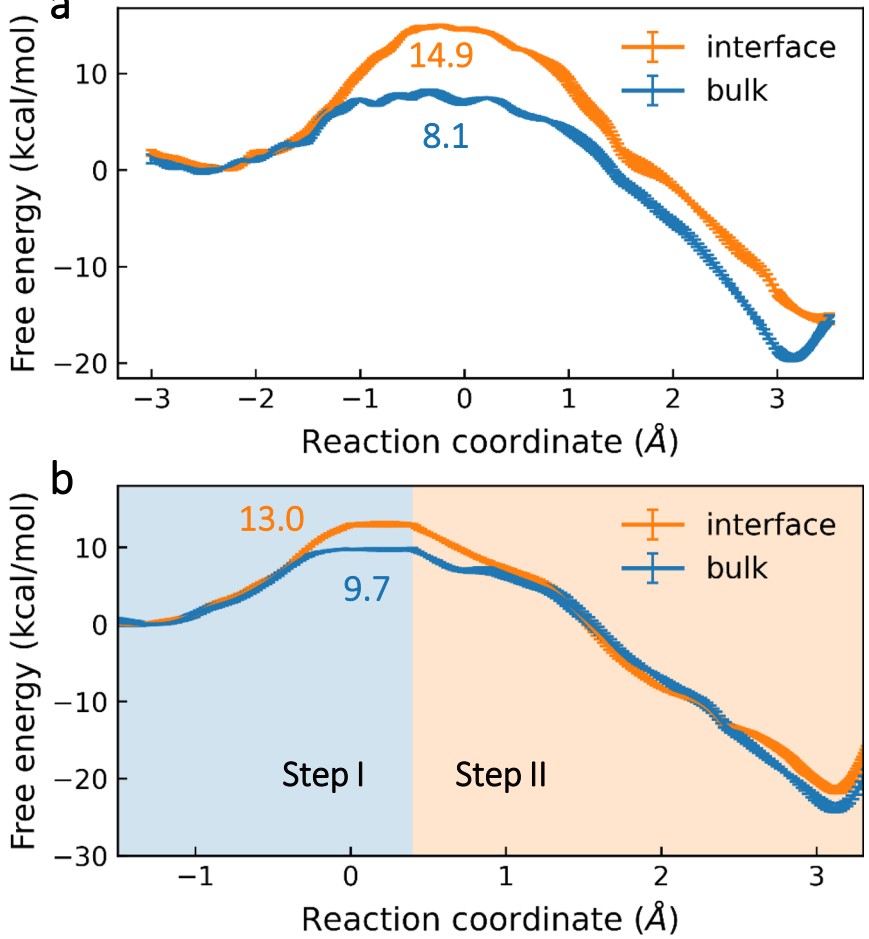

**Fig. 4 | Free energy profiles for N$_2$O$_5$ hydrolysis in liquid water via the ionic or stepwise ionic mechanisms.** Free energy profiles for N$_2$O$_5$ hydrolysis at the air–water interface (orange line) or in the bulk water (blue line) via the ionic (**a**) or stepwise ionic (**b**) mechanisms. Energy barriers are presented in kcal mol$^{-1}$. The standard deviation of the free energy of the final 8 ps of the MetaD biased QM/MM simulations is used to determine the error bars. The blue and orange shading in the background of (**b**) indicates the first and second steps, respectively, in the free energy profile for hydrolysis via stepwise ionic mechanism.

using the following equation:

$$\frac{1}{\gamma} = \frac{1}{\alpha} + \frac{\omega}{4HRT\sqrt{kD_{aq}}}\frac{1}{[coth(q) - \frac{1}{q}]} \quad (5)$$

where $\alpha$ is the mass accommodation coefficient, $\omega$ is the mean molecular speed of the gas molecule, $H$ is Henry's law constant, $RT$ is the gas constant times temperature ($T$), $k$ is the pseudo first-order rate constant, $D_{aq}$ is the diffusion coefficient, and $q$ is the reacto-diffusive parameter defined using the following equation:

$$q = R_p\sqrt{\frac{k}{D_{aq}}} = \frac{R_p}{l} \quad (6)$$

where $R_p$ is the mean particle radius, and $l$ is the reacto-diffusive length, defined as follows:

$$l = \sqrt{\frac{D_{aq}}{k}} \quad (7)$$

Experimental measurements and calculations inferred a value of $\alpha \approx 1$ for N$_2$O$_5$; thus, $1/\alpha$ was negligible. Then, the following expression

for the measured value of $\gamma$ could be obtained:

$$\gamma_{meas}(R_p) \approx \gamma_{thick}[coth(q) - \frac{1}{q}] \quad (8)$$

where $\gamma_{thick}$ is the reactive uptake coefficient in thick films and large droplets, in which the time for the gas molecule to diffuse out of the particle was much longer than the time needed for the gas molecule to chemically react within the particle:

$$\gamma_{thick} = \frac{4HRT\sqrt{kD_{aq}}}{\omega} \quad (9)$$

Previous work estimated the values of $H$, $D_{aq}$ and $\omega$ to be 3.0 M atm$^{-1}$, 10$^{-5}$ cm$^2$ s$^{-1}$ and 2.41 × 10$^4$ cm s$^{-1}$, respectively[34,35,41,42]. Our calculations at the PBE0-D3/DZVP-MOLOPT-SR level inferred bulk and interfacial hydrolysis rates of 2.3 ± 1.6 × 10$^{-3}$ and 6.3 ± 4.2 × 10$^{-7}$ ns$^{-1}$, respectively, whereas the interfacial ammonolysis rate was 151 ± 10 ns$^{-1}$. As the interfacial hydrolysis rate was four orders of magnitude slower than the bulk hydrolysis rate, it hardly contributed to the reactive uptake coefficient. Furthermore, these reaction rates implied $l_h = 26 ± 10$ nm for hydrolysis and $l_a = 0.07 ± 0.03$ nm for ammonolysis. The small value of $l_a$ indicated that the ammonolysis reaction occurred near the surface.

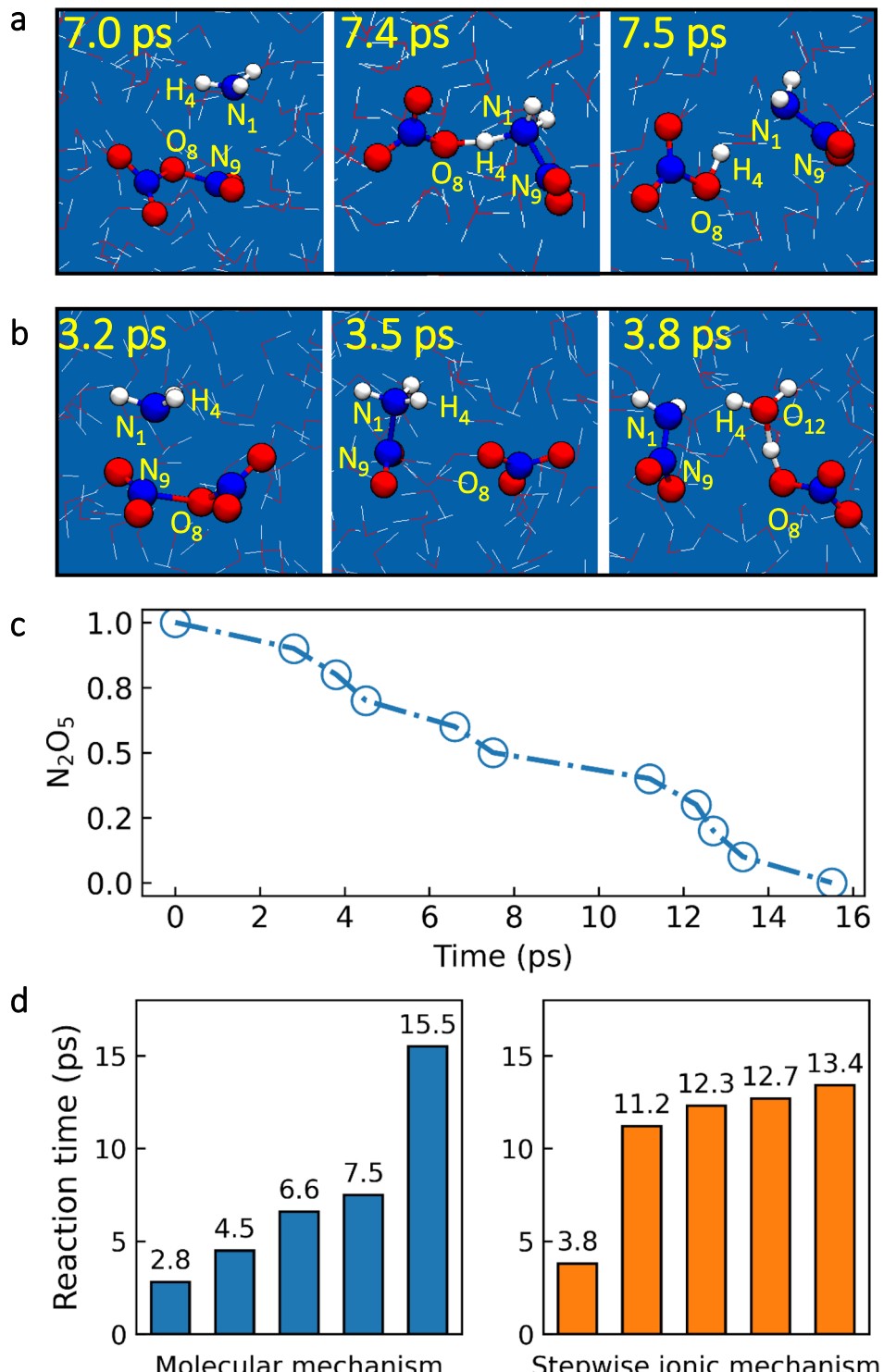

**Fig. 5 | Ultrafast ammonolysis of $N_2O_5$ at the air–water interface.** Snapshot structures for $N_2O_5$ ammonolysis at the air–water interface via the molecular (**a**) or stepwise ionic mechanism (**b**) in QM/MM MD simulations at the PBE0-D3/DZVP-MOLOPT-SR level. **c** Fraction of unreacted $N_2O_5$ as a function of simulation time in all ten independent unbiased QM/MM MD simulations at the PBE0-D3/DZVP-MOLOPT-SR level. **d** Comparison of the reaction time for $N_2O_5$ ammonolysis reactions via the molecular and stepwise ionic mechanisms.

By using the parameters above, the range of $\gamma_{thick}$ on pure water was estimated to be between 0.027 and 0.076. Figure 6 displays the predicted $\gamma$ on pure water as a function of particle radius ($R_p$). We calculated $\gamma$ by setting the hydrolysis rate equal to the upper (i.e., $3.9 \times 10^{-3}\,ns^{-1}$) and lower (i.e., $k_h = 0.7 \times 10^{-3}\,ns^{-1}$) extremes of our calculated rate. As $R_p$ increased from 40 to 130 nm, increased from 0.011–0.047 to 0.023–0.067, which was in good agreement with the experimental results, i.e., $\gamma$ of $N_2O_5$ in pure water is in the range of 0.04 and 0.06[14,43]. Unlike the reaction of $N_2O_5$ in pure water, the heterogeneous reaction of $N_2O_5$ in aqueous $NH_4HSO_4$

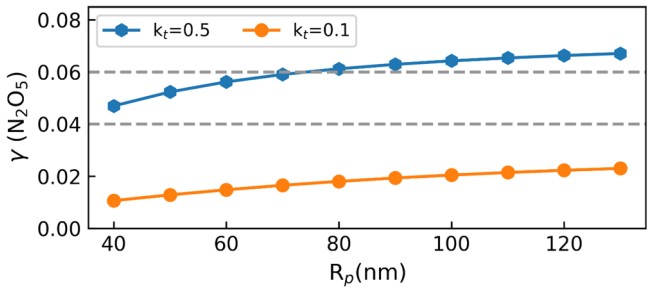

**Fig. 6 | Predicted $\gamma$ of $N_2O_5$ in pure water as a function of $R_p$.** Predicted reactive uptake coefficients of $N_2O_5$ ($\gamma$) in pure water as a function of aerosol particle radius (40–130 nm) by setting the transmission coefficient equal to the upper (i.e., $k_t = 0.5$) and lower (i.e., $k_t = 0.1$) extremes. Experimental upper and lower hydrolysis rate of $N_2O_5$ are represented by dashed line. The data for experimentally measured $\gamma$ were adapted from refs. 14,15,43.

and $(NH_4)_2SO_4$ particles was widely reported[44]. In these studies, measurements showed that $\gamma(N_2O_5)$ on aqueous $NH_4HSO_4$ and $(NH_4)_2SO_4$ particles ranged from 0.01 to 0.1. Interestingly, $\gamma$ increases with increasing RH, although the reliability of this trend at high RH is unclear[43,44], which may be due to the fact that the concentration of $NH_3$ increases with increasing RH as $NH_4^+$ is hydrolysed.

We note that Galib et al.[12] estimated a reactive uptake coefficient ($\gamma$) of 0.6 based on the widely used resistor model[39,40], which is an order of magnitude higher than experimentally derived coefficients (ranging from 0.04 to 0.06). To resolve the inconsistency, they assumed that the evaporation was barrierless and proposed that the uptake was dominated by interfacial processes. However, later MD simulations conducted by Cruzeiro et al.[13] using MB-nrg potentials challenged the conclusions of the Galib-Limmer's study. Their calculations showed that the rate of adsorption and evaporation of $N_2O_5$ at the air–water interface are 57 and 0.11 nm s⁻¹, respectively, which indicates a slow evaporation rate. Further, they found that up to 20% of the reaction occurs at the air–water interface, while most of the hydrolysis was predicted to take place in bulk water. They suggested that the reason for the disagreement may be the failure of the density functional used in the training data in Galib and Limmer's study. Indeed, our calculations using hybrid functional proved an explanation for the conflicting conclusion.

Details of the underlying mechanism for the competition between interfacial and bulk reactions of $N_2O_5$ hydrolysis and ammonolysis of $N_2O_5$ in microdroplets were deduced from a new strategy. The Bennett–Chandler method was used to quantify the rate constant for hydrolysis and ammonolysis using a high-level quantum chemical method. We estimated the bulk and interfacial hydrolysis rates to be $2.3 \pm 1.6 \times 10^{-3}$ and $6.3 \pm 4.2 \times 10^{-7}$ ns⁻¹, respectively, whereas the interfacial ammonolysis rate was $151 \pm 10$ ns⁻¹. The slow interfacial hydrolysis rate suggested that interfacial processes had negligible effect on the hydrolysis of $N_2O_5$ in liquid water. In contrast, interfacial processes dominated the ammonolysis of $N_2O_5$ in liquid water, as indicated by the high interfacial ammonolysis rate. By using the resistor model, the calculated $\gamma$ depended on the particle size of the dilute aerosol particles. As the particle radius increased from 40 to 130 nm, $\gamma$ increased from 0.011–0.047 to 0.023–0.067, which was in good agreement with the experimental results.

Atmospheric aerosols contained many chemical elements and high contents of organic substances, which could play an important role in the reactive uptake of $N_2O_5$. For example, previous studies showed that reactive uptake could be modulated with inorganic salts[15]. The strategy and framework developed here could have extensions to microdroplets with high chemical complexity, helping to provide a complete picture of the reactive uptake of $N_2O_5$ in highly complex solutions. Moreover,

systematic studies of $N_2O_5$ reactive uptake could help predict the $NO_x$ budget and the partitioning of $NO_x$ among its reservoir species.

## Methods
### Quantum chemistry calculations
We evaluated the performance characteristics of several exchange-correlation functionals, namely, generalized gradient approximation functionals (BP86[45], BLYP[46], PBE[47], revPBE[48]) and hybrid exchange-correlation functionals (B3LYP[49], PBE0[50], M06[51], B2PLYP[52]), in density functional theory (DFT) calculations of $N_2O_5$ hydrolysis and ammonolysis in the gas phase. For calculations using the revPBE functional, we used the DZVP-MOLOPT-SR[53] basis set and Goedecker–Teter–Hutter (GTH) pseudopotentials[54] and performed calculations using the CP2K 8.1 package[55]; the calculations using other functionals utilized the 6-31 + G**[53] basis set in the G09 software[56]. CCSD(T)/aug-cc-pVTZ[57,58] calculations were conducted using the ORCA 5.0[59] suite of programs. Grimme's empirical dispersion correction (D3)[60] was applied for all the DFT calculations.

### Classical MD simulations
Our liquid water model consisted of 902 water molecules that were placed in a simulation box with dimensions of $32.2 \times 32.2 \times 80$ ($x \times y \times z$) Å³, resulting in a liquid slab with two air–water interfaces. A $N_2O_5$/ $NH_3$ molecule was placed on one of the two interfaces. Classical molecular dynamics (MD) simulations combined with the umbrella sampling (US) method were performed to investigate the free energy profile of $N_2O_5$/$NH_3$ molecule transfer from the gas phase across the air–water interface into bulk water. An integration time step of 1.0 fs was used in the MD simulation. The $N_2O_5$/$NH_3$ molecule was modelled using the generalized amber force field (GAFF2)[61]. Water molecules were described by the TIP3P model[62]. We modelled nonbonding interactions using the Lennard–Jones (LJ) and Coulomb potentials. The particle–mesh Ewald summation method was used to calculate electrostatic interactions, and a real-space cut-off of 10 Å was employed for nonbonded interactions. We used the LINCS algorithm[63] to manage the bonds. The temperature was held at 300 K using a stochastic velocity rescale thermostat. Periodic boundary conditions (PBCs) were applied in all three directions. Free energies were estimated by the weighted histogram analysis method (WHAM)[64]. Classical MD simulations were performed using the GROMACS package[65].

### QM/MM simulations
The stepwise multisubphase space metadynamics (SMS-MetaD) approach[31] was combined with hybrid quantum mechanics/molecular mechanics (QM/MM) MD simulations to investigate $N_2O_5$ ammonolysis and hydrolysis at the air–water interface and inside the bulk. Specifically, we performed two steps of metadynamics (MetaD)-biased QM/MM simulations. In the first step, we aimed to identify the reaction mechanisms at the PBE-D3/DZVP-MOLOPT-SR level of theory. Numerous independent (MetaD-biased) QM/MM simulations were performed using a large QM region, which included the $N_2O_5$ molecule and all water molecules within 5 Å of any atom of the $N_2O_5$ molecule (~100 atoms). In order to prohibit the exchange of QM and MM solvent molecules, the oxygen atoms in the MM region were frozen in position, while the hydrogen atoms of water in the MM region and all atoms in the QM were free to move[66]. The collective variable (CV) selected was the distance between the N atom in the $N_2O_5$ molecule and the O atom in the water molecule ($d_{N-O}$) or the distance between the N atom in the $N_2O_5$ molecule and the N atom in the $NH_3$ molecule ($d_{N-N}$). For MetaD simulations, Gaussian hills with heights of 0.5 kcal·mol⁻¹ and sigma widths of 0.5 Å were deposited every 50 steps to efficiently search for possible reaction pathways.

In the second step, to acquire accurate free energy profiles, high-level QM/MM MD simulations with small QM regions were

conducted. The QM method at the PBE0-D3/DZVP-MOLOPT-SR level of theory was used to depict the molecules participating in chemical reactions that contained the $N_2O_5$ molecule. All atoms in the system were free to move. The identified free energy pathway was divided into multiple windows that comprised selected discrete configurations of the system. These discrete configurations were used as the initial configurations of the window to run independent MetaD-biased QM/MM MD simulations. To ensure the convergence of the free energy, the CV of each window was fully diffused in CV space at the end of the MD simulation in the second step. Finally, all subfree energy profiles were merged to obtain the free energy profile of the observed pathway. According to previous studies[67,68], the chosen CVs were linear combinations of the formation and breaking of chemical bonds involved in the reaction (Supplementary Fig. S15). Gaussian hills with a sigma width of 0.1 Å were deposited every 50 steps for each window. Two different Gaussian heights were used to accelerate convergence: a coarse Gaussian wave packet with a height of 0.25 kcal $mol^{-1}$ filled potential wells quickly first, while a fine Gaussian wave packet with a height of 0.1 kcal $mol^{-1}$ allowed the free energy to converge as smoothly as possible as the CV began to move back and forth through CV space. The convergence levels of the SMS-MetaD simulations were evaluated by CV diffusion and the variations in the free energies with time (Supplementary Figs. S16 and S17).

For the DFT calculations in the QM region, we used GTH pseudopotentials to depict the core electrons. Grimme's empirical van der Waals energy dispersion correction (D3) was used[60]. The cut-off energies for plane waves and the Gaussian basis set were set at 300 and 40 Ry, respectively. We employed the auxiliary density matrix method (ADMM)[69] to accelerate the calculations using the hybrid functional (PBE0). As in the MM model, the water molecules were described using the TIP3P model[62]. We used the real-space multigrid technique to evaluate the electrostatic interaction between the QM and MM parts[70]. All of the QM/MM MD simulations were performed in the constant volume and temperature (NVT) ensemble with the temperature maintained at 300 K using the Nosé–Hoover chain thermostat. A time step of 1.0 fs was used. The QM/MM MD simulations were conducted using the CP2K 8.1 package[55] interfaced with Plumed 2.6 software[71].

### Reporting summary

Further information on research design is available in the Nature Portfolio Reporting Summary linked to this article.

## Data availability

Computationally optimized structures and QM/MM MD input files are available at GitHub repository (https://github.com/ZhuResearch/N2O5-ammonolysis-and-hydrolysis) and Zenodo[72]. More detailed data are available from the corresponding author upon request. Source data are provided with this paper.

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

## Acknowledgements

C.Z. was supported by the National Natural Science Foundation of China (NSFC, No. 22173011).

## Author contributions

C.Z. conceived and designed the project. C. Z., J.S.F. and W.-H.F. led the project. Y.-G.F., B.T, Z.W., S.Z. discussed the implementation of hybrid functions in CP2K and SMS-MetaD methods. Y.-G.F. performed the SMS-MetaD simulation. Y.-G.F., C.Y. and L.Z. performed the DFT calculations. Y.-G.F. and C.Z. wrote the manuscript. All authors contributed to the review and editing of the manuscript and supplementary information.

## Competing interests

The authors declare no competing interests.
