## [Peer Review File · Nature Communications]

Mechanistic insight into the competition between interfacial and bulk reactions in microdroplets through N₂O₅ ammonolysis and hydrolysisREVIEWER COMMENTS

Reviewer #1 (Remarks to the Author):

In this study, Fang et al. perform ab initio molecular dynamics simulations at the level of density functional theory with hybrid density functional to simulate the hydrolysis and ammonolysis of N₂O₅ in a bulk aqueous solution and at the air-water interface. They provide their gas-phase calculations as a rationale for using hybrid density functional. They use stepwise multi-subphase space metadynamics (SMS-MetaD) to identify reaction mechanisms and estimate the free energy barriers of the two reactions.

The main contribution of this study is the observation of the stepwise ionic mechanisms in N₂O₅ hydrolysis, as well as the first observation of N₂O₅ ammonolysis under aqueous solvation at a molecular level. The calculations suggest that hydrolysis in the bulk is much more significant than at the air-water interface. However, the fast ammonolysis at the air-water interface might play an essential role in the reactive uptake of N₂O₅ by atmospheric aerosols.

The manuscript is well-written and the results are novel and interesting enough to deserve publication in Nature Communication. However, there are several issues that the authors need to address before I can recommend the publication of this work.

Major Comments:

- 1) The competition between interfacial and bulk processes is the main objective of this study. However, ammonolysis is only studied at the air-water interface. The authors might need to explain the reasons for not performing ammonolysis in bulk, as there are no bulk data to compare with the ammonolysis rate at the air-water interface.
- 2) The authors need to clarify how the air-water interface is defined, as represented by the shaded areas in Fig. S1.
- 3) Gas-phase calculations suggest that it is necessary to use hybrid functionals to obtain reaction barriers reasonably close to those computed by coupled-clusters. However, metadynamics simulations in condensed phase are performed with van der Waals corrected functionals (PBE-D3 in step 1 and PBE0-D3 in step 2). While it is clear that vdW corrections

are necessary to represent the structure of water, what can one expect about the effect of D3 correction on the reaction barriers? The functional used in the condensed phase calculations should be also benchmarked in gas phase.

4) This is related to point 3. Previous works suggest that revPBE0-D3 provides the best representation of the structure and thermodynamics of water: is there a reason why the authors chose PBE0-D3 instead? Would one expect more accurate results using revPBE0-D3?

5) The use of QM/MM in solution is complicated when part of the solvent is treated at QM level: how are the results presented in this work sensitive to the size of the QM region and/or the choice of which water molecules are treated at QM level? I would expect that some QM water molecules may leave the quantum region and be replaced by classical molecules, giving rise to a mix of QM and MM molecules, the thermodynamics of which is hard to describe (see for example works on adaptive QM/MM, e.g.

<https://pubs.acs.org/doi/10.1021/acs.jctc.6b00205>). How is this problem taken care of?

6) Also, regarding the QM/MM setup, TIP3P gives a very poor description of water. Does this choice affect the results of this study? In my opinion, the QM/MM setup needs to be validated: a possibility would be to compare the solvation shell of N2O5 at full QM (DFT) level to that obtained by QM/MM. A more extensive benchmark could involve reproducing some of the results in <https://www.nature.com/articles/s41467-022-28697-8>

7) Can the authors provide a more insightful discussion on the molecular origin of the two-step ionic mechanism in solution? In particular, is the shallow minimum in Figure 4b actually reproducible beyond the statistical errors of the simulations?

8) I recommend that raw data (input files etc.) are uploaded to a repository, such as <https://www.plumed-nest.org/>

Minor Comments:

1) It would be appropriate to add "D3" to the method used in the previous study by Galib et al., i.e., revPBE-D3/MOLOPT-DZVP, since the authors have been indicating the use of the D3 dispersion in their methods.

2) Under Fig. 1, the descriptions and the Roman numbers in the parentheses do not match.

3) Line 156, the barriers in the gas phase are determined to be 23.0 kcal/mol and 31.8 kcal/mol, respectively, as shown in Fig. 3. Why would the barriers involving additional water

molecules be higher than the gas phase?

4) The caption of Fig. S3 mentions that the stability of H_2ONO_2^+ is determined from the simulations at the PBE0-D3 level, which was inconsistent with the description in the main text. (Lines 172 to 174)

5) Line 198, a reference should be added for the experimental result.

Reviewer #2 (Remarks to the Author):

I have read the article entitled, "Mechanistic insight into the competition between interfacial and bulk reactions in microdroplets through N_2O_5 ammonolysis and hydrolysis" with great interest. Herein, the authors re investigate using molecular simulation a similar reaction to Galib- Limmer and find opposite conclusions regarding the dominant mechanism of hydrolysis and ammonolysis (not studied by Galib and Limmer). Moreover, both sets of authors use the same code (CP2K) and although they have different protocols this paper is nevertheless perplexing in its differences.

This is potentially an interesting study, but at present it offers more questions and answers.

The introduction is very nice.

However, the onus is on the present study to communicate differences to the Galib-Limmer study and in a constructive way. They do not. The only reason this referee can glean is 1) higher quality functionals (maybe), 2) QMMM vs Learned potentials, 3) Umbrella vs metaD and different reaction coordinates, 4) a different mechanism. The last is more intriguing for the Nature class journals. My gut feeling (which has been wrong) tells me that different functionals don't change the qualitative picture. The most important aspect of this study is having diffusive water. I think both the present study and the Galib-Limmer study achieve this. So questions remain about what the differences are due to. Even if one cannot completely determine where the differences come from, it is important carefully and responsibly speculate so as other researchers can test the hypothesis.

1. Galib and Limmer used umbrella sampling. This study metadynamics. MetaD is

notoriously tricky. This referee does not want to dig through the SIs to find the details. Some reasonable attempt has to be made to rule this out? If done correctly, MetaD vs Umbrella should not provide consistent results.

2. More importantly the 2D free energy surface of solvation + distance in the Galib-Limmer study seems to be a better coordinate for rates with a spot on committer distribution. The present study uses two distances and the authors should comment on this regarding the calculation of rates/committers.

3. QM/MM vs a single framework. If this is solvent mediated, then the QMMM may be problematic. QMMM is a decent framework and using QM water for the first 5 Å seems reasonable, but this may more resemble a cluster in a dielectric. Did the authors test this? Meaning--doing the reactions with the cluster in an embedded dielectric? This would be an interesting test to ensure that the QMMM is indeed mimicking the condensed phase. This needs to be discussed in the main manuscript and perhaps further convincing tests that a single QM framework vs the QMMM is equivalent for bulk studies.

4. Zero K minima are interesting with the high-level theory but we are interested in relative fluctuations. This is what drives phenomena.

5. If it turns out that the difference is solely due to the level of theory only I would be surprised. This should not change the qualitative outcome. If so, the authors must demonstrate this. The authors also must do a better job of conveying what is new and different to the Galib-Limmer study. At the moment there is nothing.

6. The non-nutritive figures of trajectories and time series and snapshots need to be changed to something that teaches us about the process.

7. Committers, etc for the dynamics would be interesting. This could lead to us understanding differences in rates.

8. If it is the two step mechanism vs. the Galib-Limmer mechanism--this needs to be clearly

demonstrated beyond being stated. This would be an interesting result and Nature worthy. One could focus the study about this and make it convincing.

At present, this referee is left guessing as to what is different. The above issues need to be addressed in a thoughtful manner. Let me reiterate that I understand that this is not a referendum on the Galib-Limmer study--Nevertheless, there needs to be something to tell the reader what is different and what can be learned that is different. Just stating that the results are different is not sufficient. Reactions at interfaces are important. Seeking consistency between the two studies and telling the community in a constructive way the potential differences (appropriately caveated) would be useful to provide a roadmap for others to continue and understand this important system.

Reviewer #4 (Remarks to the Author):

N₂O₅ is one of the key reservoir species in both stratospheric and tropospheric NO_x cycles. Its formation and removal rates thus affect both ozone levels, climate and air quality. In this manuscript, Y. G. Fang and co-workers use computational methods to assess the molecular-level details of the main sink reaction of N₂O₅: uptake into the atmospheric aqueous phase (aqueous aerosols or cloud droplets), and subsequent formation of HNO₃ (and possibly other co-products). The study reports two main findings: 1) in pure water droplets/aerosol, hydrolysis happens predominantly in the bulk phase rather than at the air-water interface as previously claimed; 2) in the presence of (some unspecified concentration of) ammonia, ammonolysis out-competes hydrolysis. While there are some issues/problems with the employed methods and the data analysis as discussed below, the overall methodology is probably sufficient at least for qualitative work (e.g. a comparison of the three channels discussed above), and the computational results are mostly reproducible with the given details (again, see below for some minor caveats). The work will thus certainly be of interest at least to atmospheric chemists working with molecular-level modelling. Assessing the broader significance of the work, for example to larger-scale modelling work concerning air quality, stratospheric ozone, or climate, is difficult to do based on the data presented so far. While the information that N₂O₅ hydrolysis happens inside rather than at the surface of droplets is interesting, it is ultimately not particularly important - the net result is still that

N₂O₅ is removed by hydrolysis. (This is reminiscent of the question of SO₃ hydrolysis, where several studies have been published demonstrating that various molecules can catalyse the hydrolysis reaction. However, their impact and relevance is limited, as hydrolysis of SO₃ to H₂SO₄ happens on sub-second timescales in any case, in almost any atmospheric conditions.) The second main result is potentially of greater relevance and impact, as it implies that the N₂O₅ removal rate might depend on the NH₃ concentration, and/or possibly the aerosol or droplet composition and pH. (Please see also comment 6 below on quantifying the dependence of the removal rate on the NH₃ concentration). However, even the “slower” quoted bulk hydrolysis rate of 2.3E-3 nanoseconds is quite fast - it implies that the lifetime of N₂O₅ in the bulk aqueous phase is less than a microsecond. Does it really matter if N₂O₅ is removed by ammonolysis in less than a nanosecond, or hydrolysis in less than a microsecond, if the net effect is in any case practically instantaneous N₂O₅ removal? To my thinking, the answer may depend on the fate of the nitramide (NH₂NO₂) product formed in the ammonolysis pathway - is this molecule expected to live long enough to affect atmospheric chemistry (or to form more long-lived products other than HNO₃)? If yes, how? A quick literature search reveals relatively few studies on nitramide oxidation, or even nitramide reactions overall, and none about actual atmospheric oxidation. Tantalisingly, some of those (see e.g. <https://pubs.acs.org/doi/10.1021/acs.est.6b04842> for an environmental chemistry study, albeit on a much larger compound) suggest N₂O as a possible product. Even limited production of N₂O from N₂O₅ in the atmospheric aqueous phase would certainly be “huge if true”. I’m certainly interested in hearing the authors’ reasoning (and possible speculations) here.

Specific comments:

1)The basis sets used in the “benchmark” geometry optimisations and energy calculations are rather modest for benchmarking purposes: 6-31+G** only has a single set of polarisation functions, while cc-pVTZ lacks any diffuse functions. Are the authors sure this is good enough for “benchmark quality”?

2)The three sets of numbers given in the first paragraph of the “Results and Discussion” section don’t appear to match each other. For example, for the 1W-1 reaction, the energy

barrier computed at “high levels of theory” is given as 14.3 to 16.8 kcal/mol. The rev-PBE barrier is then given as 8.6 kcal/mol. Finally, it is stated that this is “13.0 kcal/mol lower” than at the “benchmark” level. However, 8.6 is not 13 lower than 14.3-16.8. The same applies to all the three other reactions - the quoted difference does not match the two previous numbers. Please clarify this.

3) I don't understand how Figure 1b and d can be interpreted to suggest that “revPBE performed the best”. Especially for the ammonolysis reaction, revPBE performs very poorly, much worse than B3LYP, PBE0 or M02X, all of which are methods with roughly similar computational expense. (B2PLYP is much more expensive.) Is the argument here that revPBE is somehow a “low-level” method, while e.g. B3LYP is “high level”? Are the red columns in the figure supposed to be “low level”, and the blue columns “high”? This division seems arbitrary and even incorrect, and even if it were true, revPBE is not really much better than the other “red” methods (which are all atrociously bad).

4) Do any of the entries in Figures 1b and d correspond to the methods actually used in the QM/MM simulations (BPE-D3/DZVP-MOLOPT-SR and PBE0-D3/DZVP, according to the methods - section)? If yes, please label it accordingly - if no, please redo the calculations and include this method in the figure!

5) Please explain from which data the transition-state theory rates are computed. I assume it is the PBE0-D3/DZVP data in the “high-level” QM/MM simulations? Given the rather large differences between PBE and CCSD(T) in Figures 1b and 1d, how reliable should the absolute rates be considered? Should they perhaps be reduced by some correction factor proportional to $\exp(-dE/RT)$, where dE is the difference in barrier heights between the methods? (Note that the answer to the previous question may render this question either more or less relevant.)

6) The hydrolysis rate is expressed in terms of a unimolecular rate coefficient (in units of 1/ns). While the reaction between N_2O_5 and H_2O is of course bimolecular, in an aqueous-phase (bulk or interface) context this is appropriate: the concentration of water is after all close to constant (even in relatively concentrated solutions), and it makes sense to implicitly

include it in the rate coefficient. However, for ammonolysis this is not the case - the rate at which N_2O_5 is ammonolysed will certainly depend on the ammonia concentration, so giving one single pseudo-unimolecular number is meaningless. Please provide instead a figure or table of the pseudo-unimolecular ammonolysis rate as a function of the liquid-phase ammonia concentration, or even the gas-phase ammonia concentration. The latter actually becomes an interesting exercise, as the $[\text{NH}_3]_{\text{aq}}$ depends not only on p_{NH_3} but also on pH - unless also NH_4^+ ions can ammonolyse N_2O_5 ? As a side note, the results would imply a possible pH - dependence of the N_2O_5 uptake coefficient, which would be intriguing, albeit with two caveats: 1) also the hydrolysis lifetime is so short that this dependence may not matter; and 2) just like in the case of SO_3 mentioned above, it may well turn out that aqueous-phase N_2O_5 destruction is catalysed by many different species, including both acidic and basic molecules. (The latter argument is certainly beyond the scope of the present manuscript, I'm just raising the issue as something the authors might want to look at later).

7) The authors are apparently comparing their pure-water uptake coefficients to measurements done for ammonium sulfate and bisulfate particles (discussion around Figure 6). However, their central argument in this study is that ammonia affects the uptake process! Does this not invalidate the comparison?

8) Please explain a bit more what the transmission coefficient $k(t)$ in 1 accounts for. Also, what does the "(t)" notation here mean - how is the coefficient time-dependent?

Response to Reviewers

Reviewer #1

In this study, Fang et al. perform ab initio molecular dynamics simulations at the level of density functional theory with hybrid density functional to simulate the hydrolysis and ammonolysis of N_2O_5 in a bulk aqueous solution and at the air-water interface. They provide their gas-phase calculations as a rationale for using hybrid density functional. They use stepwise multi-subphase space metadynamics (SMS-MetaD) to identify reaction mechanisms and estimate the free energy barriers of the two reactions.

The main contribution of this study is the observation of the stepwise ionic mechanisms in N_2O_5 hydrolysis, as well as the first observation of N_2O_5 ammonolysis under aqueous solvation at a molecular level. The calculations suggest that hydrolysis in the bulk is much more significant than at the air-water interface. However, the fast ammonolysis at the air-water interface might play an essential role in the reactive uptake of N_2O_5 by atmospheric aerosols.

The manuscript is well-written and the results are novel and interesting enough to deserve publication in Nature Communication. However, there are several issues that the authors need to address before I can recommend the publication of this work.

Author response: We appreciate the reviewer for the positive evaluation of our work and the constructive comments that helped to improve the clarity of the manuscript. We have carefully considered these suggestions and are committed to addressing the following issues to meet the standards of Nature Communications.

Reviewer #1 Major Comments:

1) The competition between interfacial and bulk processes is the main objective of this study. However, ammonolysis is only studied at the air-water interface. The authors might need to explain the reasons for not performing ammonolysis in bulk, as there are no bulk data to compare with the ammonolysis rate at the air-water interface.

Author response: We thank the reviewer for the comment. In this study, we did not study the ammonolysis in bulk for two reasons:

- 1). As shown in supplementary Fig. 1, both N_2O_5 and NH_3 have interfacial affinity properties, which means that they are prone to collisions and reactions at the air-water interface.
- 2). The kinetics of N_2O_5 ammonolysis reaction are exceptionally rapid, resulting in the N_2O_5 ammonolysis reaction occurring before N_2O_5 diffuses into the bulk phase. Indeed, the reacto-diffusive length is given by the following equation:

$$l = \sqrt{\frac{D_{aq}}{k}}$$

where D_{aq} is the diffusion coefficient and k is the pseudo first-order rate constant. Our calculations show that the lifetime of ammonolysis is ~ 9 ps. On the other hand, the diffusion coefficient is about 10^{-5} cm^2/s , and the reacto-diffusive length is calculated to be about 0.09 nm. Therefore, the reaction should be completed at the interface. Based on the reviewer's comment, we have made corresponding changes in the revised manuscript.

- 2) The authors need to clarify how the air-water interface is defined, as represented by the shaded areas in Fig. S1.

Author response: We thank the referee for the comment. In this study, we use the widely accepted 10-90 thickness to define the air-water interface. Specifically, the air-water interface is defined as the region between 90 and 10% of the bulk density, which has been widely adopted (*J. Chem. Phys.* 102, 4574-4583 (1995)). Based on the reviewer's comment, we have clarified how to define the air-water interface in the revised manuscript.

- 3) Gas-phase calculations suggest that it is necessary to use hybrid functionals to obtain reaction barriers reasonably close to those computed by coupled-clusters. However, metadynamics simulations in condensed phase are performed with van der Waals corrected functionals (PBE-D3 in step 1 and PBE0-D3 in step 2). While it is clear that vdW corrections are necessary to represent the structure of water, what can one expect about the effect of D3 correction on the reaction barriers? The functional used in the condensed phase calculations should be also benchmarked in gas phase.

Author response: We thank the referee for this suggestion. Following the suggestion, we investigate the effect of D3 correction on the reaction barriers. Fig. R1 shows the reaction energy barriers of N_2O_5 hydrolysis for different mechanisms in the gas phase calculated using

the PBE0 functional with and without D3 correction. Apparently, the D3 correction has little effect on the hydrolysis reaction of N_2O_5 in the gas phase.

Fig. R1. Reaction energy barriers of N_2O_5 hydrolysis for different mechanisms in the gas phase calculated using the PBE0 functional with and without Grimme's D3 correction.

4) This is related to point 3. Previous works suggest that revPBE0-D3 provides the best representation of the structure and thermodynamics of water: is there a reason why the authors chose PBE0-D3 instead? Would one expect more accurate results using revPBE0-D3?

Author response: We thank the referee for the comment. The selection of the PBE0-D3 functional is based on a balance between computational efficiency and accuracy. We calculated the reaction energy barriers of N_2O_5 hydrolysis for different mechanisms in the gas phase using revPBE0-D3 and PBE0-D3 functionals. As shown in Fig. R2, the two quantum chemical methods give almost identical energy barriers (the difference in energy barriers is less than 10%). On the other hand, computational efficiency is also crucial. According to our test, PBE0-D3 is 2-3 times more computationally efficient than revPBE0-D3. Therefore, although revPBE0-D3 may perform better in characterizing structural and thermodynamic properties of water, we chose PBE0-D3 as the more suitable option for our research.

Fig. R2. Reaction energy barriers of N_2O_5 hydrolysis for different mechanisms in the gas phase calculated using revPBE0-D3 and PBE0-D3 functionals.

5) The use of QM/MM in solution is complicated when part of the solvent is treated at QM level: how are the results presented in this work sensitive to the size of the QM region and/or the choice of which water molecules are treated at QM level? I would expect that some QM water molecules may leave the quantum region and be replaced by classical molecules, giving rise to a mix of QM and MM molecules, the thermodynamics of which is hard to describe (see for example works on adaptive QM/MM, e.g. <https://pubs.acs.org/doi/10.1021/acs.jctc.6b00205>). How is this problem taken care of?

Author response: We thank the referee for the comment. Standard conventional AIMD simulations using high-level quantum chemical methods (e.g. PBE0-D3) to explore the reaction of N_2O_5 at the air–water interface and inside the bulk are virtually impossible. To overcome the difficulties, we performed QM/MM MD simulations using our developed SMS-MetaD method¹ to simulate the reaction of N_2O_5 in this study. We performed two-step MetaD simulations, In the first step, MetaD-biased QM/MM MD simulations with a large QM region, which comprises ~100 atoms, were performed at the PBE-D3/DZVP level of theory to identify free energy pathways (FEPs). Unlike the adaptive QM/MM simulations, the oxygen atoms in the MM region were frozen in position to prohibit the exchange of QM and MM solvent molecules, while the hydrogen atoms of water in MM region and all atoms in the QM region were free to move. To ensure that the oxygen atom positions in the MM region were reasonable, we ran 2.0 ns classical MD simulations prior to the QMMM dynamics simulations. A similar strategy has been adopted previously.²⁻³ In the second step, high-level QM/MM MD simulations with a small QM region. The QM method at the PBE0-D3/DZVP level of theory was used to depict the molecules participating in chemical reactions that contained the N_2O_5

molecule, and all atoms in the MM region were free to move. Different with standard MetaD method, the identified FEP was divided into a series of independent subphase spaces and the accurate free energy landscape for each subphase space was separately calculated. In the revised manuscript, we have made relevant changes and cited the paper.

Reference:

1. Fang Y-G, et al. Efficient exploration of complex free energy landscapes by stepwise multi-subphase space metadynamics. *J. Chem. Phys.* 157, 214111 (2022).
2. Wang SH, et al. Development of Semiempirical Models for Proton Transfer Reactions in Water. *J. Chem. Theory Comput.* 10, 2881-2890 (2014).
3. Zhu CQ, et al. Hydration, Solvation, and Isomerization of Methylglyoxal at the Air/Water Interface: New Mechanistic Pathways. *J. Am. Chem. Soc.* 142, 5574-5582 (2020).

6) Also, regarding the QM/MM setup, TIP3P gives a very poor description of water. Does this choice affect the results of this study? In my opinion, the QM/MM setup needs to be validated: a possibility would be to compare the solvation shell of N₂O₅ at full QM (DFT) level to that obtained by QM/MM. A more extensive benchmark could involve reproducing some of the results in <https://www.nature.com/articles/s41467-022-28697-8>

Author response: We thank the referee for the comment. Regarding the selection of the TIP3P water model, we would like to emphasize that the TIP3P water model is indeed widely used in many QM/MM applications and has demonstrated its ability to yield accurate results in a variety of situations.¹⁻⁶ Following the referee's suggestion, we analyzed the solvation shell of N₂O₅ by computing the nitrogen-oxygen radial distribution function, which was obtained by the QM/MM method (revPBE-D3/DZVP//TIP3P). As shown in Fig. R3a and 3b, although the height of the first peak obtained by the QM/MM method is slightly lower than that obtained by AIMD simulations or neural network potential based MD simulations, the QM/MM method can closely reproduce the location of the peaks obtained by full DFT calculations. Furthermore, we combined the SMS-MetaD method with the QM/MM (revPBE-D3/DZVP//TIP3P) MD simulation to investigate the hydrolysis of N₂O₅ in bulk water. As shown in Fig. 3c and 3d, the free energy barrier for the reaction is 4.6 ± 0.2 kcal/mol, in good agreement with that obtained by Galib and Limmer using a neural network model (~ 3.8 kcal/mol) and our AIMD simulation (5.8 ± 0.3 kcal/mol).

In addition, we investigated the thermodynamics of N₂O₅ solvation in liquid water using the QM/MM method (revPBE-D3/DZVP//TIP3P). Fig. R3e depicts the free energy profile for

moving a gaseous N_2O_5 into liquid water. The free energy exhibits a global minimum centered at the air-water interface, and this global minimum relative to the gas phase ($\Delta F_a = -3.5$ kcal/mol) corresponds to an interfacial adsorption free energy. Moreover, the difference between the free energy of N_2O_5 in bulk water and that in gas phase ($\Delta F_s = -1.5$ kcal/mol) is defined as the solvation free energy for the gaseous N_2O_5 . Apparently, the calculated ΔF_a agrees very well with the ΔF_a obtained using the many body potential ($\Delta F_a = -3.7$ kcal/mol), which was parameterized from coupled-cluster calculations, and outperforms that calculated using the neural network potential ($\Delta F_a = -2.7$ kcal/mol). For ΔF_a , although the value calculated by QM/MM is somewhat higher than the one calculated using the many body potential ($\Delta F_s = -2.7$ kcal/mol), it is almost the same as the one calculated based on the neural network potential ($\Delta F_a = -1.5$ kcal/mol). These results validate the effectiveness of the QM/MM method in studying the N_2O_5 reaction.

Fig. R3. Radical distribution function of $\text{N}(\text{N}_2\text{O}_5)\text{-O}(\text{H}_2\text{O})$ for solvated N_2O_5 in bulk water using

the QM/MM method (revPBE-D3/DZVP//TIP3P) (a), the neural network potential (dashed lines) and full DFT method (red line) as studied by Galib and Limmer (b). Free energy profiles of N_2O_5 hydrolysis in bulk water via the ionic mechanism obtained by (c) QM(revPBE-D3)/MM MD simulations and (d) AIMD (revPBE-D3) simulations. Free energy profile for moving a gaseous N_2O_5 into liquid water using the QM/MM method (revPBE-D3/DZVP//TIP3P) (e) and the neural network potential (f) as studied by Galib and Limmer.

Reference:

1. Ishida T, Parks JM, Smith JC. Insight into the Catalytic Mechanism of GH11 Xylanase: Computational Analysis of Substrate Distortion Based on a Neutron Structure. *J. Am. Chem. Soc.* 142, 17966-17980 (2020).
2. Ruiz-Lopez MF, et al. Tight electrostatic regulation of the OH production rate from the photolysis of hydrogen peroxide adsorbed on surfaces. *Proc. Natl. Acad. Sci. U. S. A.* 118, (2021).
3. Anglada JM, et al. Reactivity of Undissociated Molecular Nitric Acid at the Air-Water Interface. *J. Am. Chem. Soc.* 143, 453-462 (2021).
4. Zhou SY, et al. Solvation Induction of Free Energy Barriers of Decarboxylation Reactions in Aqueous Solution from Dual-Level QM/MM Simulations. *JACS Au* 1, 233-244 (2021).
5. Morais MAB, et al. Two distinct catalytic pathways for GH43 xylanolytic enzymes unveiled by X-ray and QM/MM simulations. *Nat. Commun.* 12, (2021).
6. Martins-Costa MTC, Ruiz-López MF. Electrostatics and Chemical Reactivity at the Air-Water Interface. *J. Am. Chem. Soc.*, 145, 1400–1406, (2023).
7. Galib M, Limmer DT. Reactive uptake of N_2O_5 by atmospheric aerosol is dominated by interfacial processes. *Science* 371, 921-925 (2021).

7) Can the authors provide a more insightful discussion on the molecular origin of the two-step ionic mechanism in solution? In particular, is the shallow minimum in Figure 4b actually reproducible beyond the statistical errors of the simulations?

Author response: We thank the referee for the comment. Fig. R4 shows the Mulliken partial charges on the NO_2 and NO_3 fragments of N_2O_5 when N_2O_5 is at the air-water interface, in the gas phase, and in bulk water, as obtained from previous literature (*Phys. Chem. Chem. Phys.* 20, 17961-17976 (2018)). It is clear that there are some charge fluctuations within the N_2O_5 molecule in all three cases. Note that the charge separation of N_2O_5 at the air-water interface and in bulk water is greater than that of N_2O_5 in the gas phase. In other words, the N atom in

the NO_2 fragment of N_2O_5 at the air-water interface and in bulk water is more positively charged than N_2O_5 in the gas phase. On the other hand, the O atom in water molecule is an electron-rich atom. Thus, the intermediate H_2ONO_2^+ ion can be formed via the interaction between the N and O atoms of water molecules.

Fig. R4 Charge distribution of N_2O_5 at the air-water interface, in the gas phase and in bulk water. The top panel shows the charge on the NO_2 (red) and NO_3 (blue) fragments of N_2O_5 . The bottom panel shows the probability distribution for the NO_3 fragment to have a certain partial charge. The probability distribution for NO_2 approximately mirrors that of NO_3 and is not shown. The blue curve presents the distribution while the dashed black line presents two standard deviations from the mean distribution. The figure is taken from previous literature. (*Phys. Chem. Chem. Phys.* 20, 17961-17976 (2018))

To address the referee's concern about the shallow minimum, we performed seven independent unbiased QM/MM MD simulations of H_2ONO_2^+ at the air-water interface and in bulk water, respectively. Typical snapshot structures along the reaction taken from a QM/MM MD simulation are shown in Fig. R5 (a). Fig. R5 (b) shows the fraction of H_2ONO_2^+ unreacted versus simulation time. The average lifetimes are 0.78 ps and 0.31 ps for H_2ONO_2^+ at the air-water interface and in bulk water, respectively, which are much longer than the average proton transfer time (less than 0.1 ps). (*J. Am. Chem. Soc.* 145, 10159-10166 (2023)) The relatively long lifetime of H_2ONO_2^+ indicates that it is an intermediate.

Fig. R5. (a) Typical snapshot structures taken from a QM/MM MD simulation of H_2ONO_2^+ in bulk water. (b) Fraction of unreacted intermediate H_2ONO_2^+ at the air-water interface or in bulk water as a function of simulation time.

8) I recommend that raw data (input files etc.) are uploaded to a repository, such as <https://www.plumed-nest.org/>

Author response: We thank the referee for the suggestion. Relevant input files have been uploaded to <https://github.com/FANGYEGUANG9527/N2O5-hydrolysis-and-ammonolysis>.

Minor Comments:

1) It would be appropriate to add “D3” to the method used in the previous study by Galib et al., i.e., revPBE-D3/MOLOPT-DZVP, since the authors have been indicating the use of the D3 dispersion in their methods.

Author response: We thank the referee for the suggestion. We have added “D3” to the method in the work conducted by Galib et al. in the revised manuscript.

2) Under Fig. 1, the descriptions and the Roman numbers in the parentheses do not match.

Author response: Thanks! We have corrected the descriptions under Fig. 1 accordingly in the revised manuscript.

3) Line 156, the barriers in the gas phase are determined to be 23.0 kcal/mol and 31.8 kcal/mol, respectively, as shown in Fig. 3. Why would the barriers involving additional water

molecules be higher than the gas phase?

Author response: We thank the referee for the comment and apologize for the lack of clarity. In the gas phase, the involvement of additional water molecules will result in lower barriers (Fig. 1). However, our calculations reveal that the barriers for the reaction of N_2O_5 with water monomer, water dimer or water trimer at the air-water interface via the molecular mechanism are in the order water trimer > water dimer > water monomer. We have made corresponding changes in the revised manuscript:

“Surprisingly, unlike the gas phase reactions, the energy barrier for the reaction of N_2O_5 with water dimer or water trimer at the air-water interface is higher than that for the reaction of N_2O_5 with water monomer, which is ~15.9 kcal/mol or ~19.3 kcal/mol, respectively.”

4) The caption of Fig. S3 mentions that the stability of $H_2ONO_2^+$ is determined from the simulations at the PBE0-D3 level, which was inconsistent with the description in the main text. (Lines 172 to 174)

Author response: Thanks! We have made corresponding changes in the revised manuscript.

5) Line 198, a reference should be added for the experimental result.

Author response: Thanks! We have added the reference in the revised manuscript.

Reviewer #2

I have read the article entitled, "Mechanistic insight into the competition between interfacial and bulk reactions in microdroplets through N₂O₅ ammonolysis and hydrolysis" with great interest. Herein, the authors re investigate using molecular simulation a similar reaction to Galib-Limmer and find opposite conclusions regarding the dominant mechanism of hydrolysis and ammonolysis (not studied by Galib and Limmer). Moreover, both sets of authors use the same code (CP2K) and although they have different protocols this paper is nevertheless perplexing in its differences.

This is potentially an interesting study, but at present it offers more questions and answers.

The introduction is very nice.

However, the onus is on the present study to communicate differences to the Galib-Limmer study and in a constructive way. They do not. The only reason this referee can glean is 1) higher quality functionals (maybe), 2) QMMM vs Learned potentials, 3) Umbrella vs metaD and different reaction coordinates, 4) a different mechanism. The last is more intriguing for the Nature class journals. My gut feeling (which has been wrong) tells me that different functionals don't change the qualitative picture. The most important aspect of this study is having diffusive water. I think both the present study and the Galib-Limmer study achieve this. So questions remain about what the differences are due to. Even if one cannot completely determine where the differences come from, it is important carefully and responsibly speculate so as other researchers can test the hypothesis.

Author response: We appreciate your interest in our work and your valuable feedback. We acknowledge your comments regarding the differences between our study and those of Galib and Limmer.

We note that Galib *et al.* estimated a reactive uptake coefficient (γ) of 0.6 based on the widely used resistor model, which is an order of magnitude higher than that obtained from experiments (ranging from 0.04 to 0.06). To resolve the inconsistency, they proposed that high evaporation rate played a key role. (*Science* 371, 921-925 (2021)) However, later MD simulations (*Nat. Commun.* 13, 1266 (2022)) using chemically accurate many-body potentials

challenged the conclusions of the Galib-Limmer study. Note that Galib and Limmer are also authors of the paper. They suggested that “The disagreement with respect to the neural network model could likely be a failure of the density functional used in the training data”. This inspired us to use higher quality functionals to investigate the reactive uptake of N_2O_5 by atmospheric aerosol.

To address the concerns of the referee, further simulations were conducted to investigate the reasons behind the differences, which are discussed in the revised manuscript.

Reviewer #2 Major Comments:

1) Galib and Limmer used umbrella sampling. This study metadynamics. MetaD is notoriously tricky. This referee does not want to dig through the SIs to find the details. Some reasonable attempt has to be made to rule this out? If done correctly, MetaD vs Umbrella should not?? provide consistent results.

Author response: We thank the referee for the comment. We agree with the referee that metaD is tricky. In this study, we used our own developed SMS-MetaD method, rather than the standard metaD method, to investigate the reactions at the air-water interface and in bulk water. Note that the effectiveness of SMS-MetaD has been validated by studies of formaldehyde hydrolysis in the gas phase and at the air-water interface. (*J. Chem. Phys.* 157, 214111 (2022))

To further validate the SMS-MetaD method, we combined it with *ab initio* MD simulations to simulate the hydrolysis of N_2O_5 in bulk water at the revPBE-D3 level of theory. We use the revPBE-D3 here because the neural network potential used in the Galib-Limmer's study was fitted based on the revPBE functional. The bulk system contained one N_2O_5 molecule solvated by 192 water molecules in a $18.06 \times 18.06 \times 18.06 \text{ \AA}^3$ box with periodic boundary conditions in all three dimensions. Fig. R6a shows the collective variable (CV) used in the simulation. Fig. 6b displays the free energy profile for the hydrolysis of N_2O_5 as a function of the CV. The free energy barrier for the reaction is $5.8 \pm 0.3 \text{ kcal/mol}$, which is in agreement with that obtained by Galib and Limmer using umbrella sampling method and the neural network model ($\sim 3.8 \text{ kcal/mol}$). In addition, we have analyzed the nitrogen-nitrogen distance (R) and the coordination number between the nitrogen atoms in N_2O_5 and the surrounding water molecules (n_w). The results show that the transition state is located at $n_w = 0.5 \pm 0.1$ and $R = 3.15 \pm 0.15 \text{ \AA}$, which is also in agreement with the results of Galib and Limmer ($n_w = 0.4$ and $R = 3.1 \text{ \AA}$). Overall, the SMS-MetaD method and the umbrella sampling method yielded

consistent results.

Fig. R6. (a) Collective variable (CV) used in MetaD-biased AIMD simulation of N_2O_5 hydrolysis in bulk water. (b) Free energy profile for N_2O_5 hydrolysis in bulk water via the ionic mechanism simulated at the revPBE-D3 level of theory.

2) More importantly the 2D free energy surface of solvation + distance in the Galib-Limmer study seems to be a better coordinate for rates with a spot on committer distribution. The present study uses two distances and the authors should comment on this regarding the calculation of rates/committers.

Author response: We thank the referee for the comment. We agree with the referee that solvation and distance are widely used CVs for the reactions at the air-water interface or in bulk water. However, it has been widely reported that multiple reaction pathways may exist in hydrolysis in the gas, in bulk water and at the air-water interface.¹⁻⁴ For example, unbiased AIMD simulations have shown that the reaction between CH_2OO and water at the air/water interface can occur via multiple reaction pathways involving different numbers of reactive water molecules.¹ To distinguish between different mechanisms, all the bond distances involved in the reaction are linearly combined to form a CV, depending on whether the bond is formed or broken. Moreover, such a CV is able to correctly distinguish reactants, transition states and products. Note that a similar CV selection strategy is widely used.⁵⁻⁷

Based on the referee's comment, we combined the SMS-MetaD method with AIMD simulation to investigate the hydrolysis of N_2O_5 in bulk water using the same functional as used in the study by Galib and Limmer. Fig. R6a shows the collective variable (CV) used in the simulation. The results of our simulations are in good agreement with those obtained by

Galib and Limmer using solvation and distance as CVs. See our response above for details.

Reference:

1. Zhu CQ, et al. New Mechanistic Pathways for Criegee-Water Chemistry at the Air/Water Interface. *J. Am. Chem. Soc.* 138, 11164-11169 (2016).
2. Anglada JM, Sole A. Impact of the water dimer on the atmospheric reactivity of carbonyl oxides. *Phys. Chem. Chem. Phys.* 18, 17698-17712 (2016).
3. Liu JF, Liu YQ, Yang JR, Zeng XC, He X. Directional Proton Transfer in the Reaction of the Simplest Criegee Intermediate with Water Involving the Formation of Transient H_3O^+ . *J. Phys. Chem. Lett.* 12, 3379-3386 (2021).
4. Zhu CQ, et al. Hydration, Solvation, and Isomerization of Methylglyoxal at the Air/Water Interface: New Mechanistic Pathways. *J. Am. Chem. Soc.* 142, 5574-5582 (2020).
5. Piccini G, et al. Variational Flooding Study of a SN2 Reaction. *J. Phys. Chem. Lett.* 8, 580-583 (2017).
6. Piccini G, et al. Metadynamics with Discriminants: A Tool for Understanding Chemistry. *J. Chem. Theory Comput.* 14, 5040-5044 (2018).
7. Chen M, et al. Hydroxide diffuses slower than hydronium in water because its solvated structure inhibits correlated proton transfer. *Nat. Chem.* 10, 413-419 (2018).

3) QM/MM vs a single framework. If this is solvent mediated, then the QMMM may be problematic. QMMM is a decent framework and using QM water for the first 5 Å seems reasonable, but this may more resemble a cluster in a dielectric. Did the authors test this? Meaning--doing the reactions with the cluster in an embedded dielectric? This would be an interesting test to ensure that the QMMM is indeed mimicking the condensed phase. This needs to be discussed in the main manuscript and perhaps further convincing tests that the single QM framework vs the QMMM is equivalent for bulk studies.

Author response: We thank the referee for the comment. To verify the validity of the QM/MM framework, we combined the SMS-MetaD method with QM/MM MD simulations to study the hydrolysis of N_2O_5 in bulk water. We used the revPBE-D3 to depict the molecules participating in chemical reactions, which contained the N_2O_5 molecule, and all atoms in the MM region were free to move. We used the revPBE-D3 here because the neural network potential used in the Galib-Limmer's study was fitted based on the revPBE-D3 functional. As shown in Fig. R7, the free energy barrier for this reaction is 4.6 ± 0.2 kcal/mol, which is in good agreement with that obtained by Galib and Limmer using a neural network model (~ 3.8 kcal/mol) and

with our AIMD simulations (5.8 ± 0.3 kcal/mol). In addition, we also analyzed R and n_w in our QM/MM MD simulations. The results show that the transition state is located at $n_w = 0.38 \pm 0.09$ and $R = 3.26 \pm 0.22$ Å, which is also in good agreement with the results of Galib and Limmer ($n_w = 0.4$ and $R = 3.1$ Å) as well as our AIMD simulations ($n_w = 0.5 \pm 0.1$ and $R = 3.15 \pm 0.15$ Å).

Fig. R7. Free energy profile for N₂O₅ hydrolysis in bulk water via the ionic mechanism. MetaD-biased QM/MM MD simulation was performed. The QM part was simulated at the revPBE-D3 level of theory.

4) Zero K minima are interesting with the high-level theory but we are interested in relative fluctuations. This is what drives phenomena.

Author response: We thank the referee for the comment.

5) If it turns out that the difference is solely due to the level of theory only I would be surprised. This should not change the qualitative outcome. If so, the authors must demonstrate this. The authors also must do a better job of conveying what is new and different to the Galib-Limmer study. At the moment there is nothing.

Author response: We thank the referee for the comment. As mentioned above, our tests verified the validity of the computational chemistry strategy we developed (see above for details). Indeed, it is not surprising that the difference between our calculations and Galib-Limmer's study lies mainly in the level of theory used. We note that Galib *et al.* estimated a reactive uptake coefficient (γ) of 0.6 based the widely used resistor model, which is an order of magnitude higher than the experimentally derived coefficients (ranging from 0.04 to 0.06). To resolve this inconsistency, they assumed that evaporation was barrierless and proposed that a high evaporation rate was the key factor. However, later MD simulations using MB-nrg potentials challenged the conclusions of the Galib-Limmer's study. Note that Galib and Limmer are also authors of the paper. The MB-nrg model was demonstrated to yield

comparable accuracy with respect to the coupled cluster reference data it was parameterized. Their calculations show that the evaporation rate is very slow compared to the adsorption rate with “*corresponding higher accommodation coefficient relative to the previous neural network model study*” (Galib-Limmer's study). In addition, by solving the reaction-diffusion equation, they found that interfacial reactivity “*accounts for at most 20%*”, whereas Galib-Limmer's study estimated that almost 100% reactions occurred at the air-water interface. They suggested that “***The disagreement with respect to the neural network model could likely be a failure of the density functional used in the training data***”. This inspired us to use higher quality functionals to investigate the reactive uptake of N_2O_5 by atmospheric aerosol.

In this work, we have developed a strategy to simulate reactions at the air-water interface and in bulk water using high quality functionals. Our approach successfully identified different mechanisms of N_2O_5 hydrolysis at the air-water interface and in bulk water, such as molecular mechanisms and ionic mechanisms. Additionally, we confirmed for the first time the existence of a stepwise ionic mechanism. Using SMS-MetaD method combined with QM/MM MD simulations, we obtained the free energy profiles for N_2O_5 hydrolysis via different mechanisms. In agreement with Cruzeiro et al. but different from Galib-Limmer's study, most of the hydrolysis is predicted to take place in bulk water. Using the resistor model, we predicted the variation of the reaction uptake coefficients γ of N_2O_5 in pure water as a function of aerosol particle radius (R_p). As the particle radius increases from 40 nm to 130 nm, γ increases from 0.011~0.047 to 0.023~0.067, which is in good agreement with experimental results. Finally, we have investigated for the first time the ammonolysis of N_2O_5 in liquid water, and our calculations show that, unlike the hydrolysis process, the interfacial process dominates the ammonolysis process due to the high interfacial ammonolysis rate. We have discussed the novelties and differences of our work compared to the Galib-Limmer's study in the revised manuscript.

6) The non-nutritive figures of trajectories and time series and snapshots need to be changed to something that teaches us about the process.

Author response: We thank the referee for the comment. We have modified Fig. 2 and Fig. 5 in the revised manuscript to improve its clarity.

7) Committers, etc for the dynamics would be interesting. This could lead to us understanding differences in rates.

Author response: We thank the referee for the suggestion. As the ionic mechanism

governed the N_2O_5 hydrolysis reaction in liquid water, we have analysed transition states for N_2O_5 hydrolysis via ionic mechanism. Specifically, we investigated 30 configurations belonging to the constrained ensemble with reaction coordinate (RC) ranges from -0.025 to 0.025. The committers of configurations are narrowly distributed around 0.5 (Fig. R8), indicating that the transition state criterion used is good.

Fig. R8. Distribution $p(P_B)$ of the probability for relaxing to the product within the constrained ensemble with $\text{RC} \in [-0.025, 0.025]$.

8) If it is the two step mechanism vs. the Galib-Limmer mechanism--this needs to be clearly demonstrated beyond being stated. This would be an interesting result and Nature worthy. One could focus the study about this and make it convincing.

Author response: We thank the referee for the comment. The stepwise ionic hydrolysis mechanism is indeed intriguing and has never been reported in the literature before. However, the difference between our calculations and Galib-Limmer's study lies mainly in the level of theory used. In the study, we obtained the free energy profiles for N_2O_5 hydrolysis via different mechanisms and showed that the calculated free energy barrier at the air-water interface and in bulk water are in the order of ionic mechanism > molecular mechanism \approx stepwise ionic mechanism and molecular mechanism > stepwise ionic mechanism > ionic mechanism, respectively. These calculations suggest that stepwise ionic and molecular mechanisms play a major role in the N_2O_5 hydrolysis reaction at the air-water interface, whereas the ionic mechanism governed the N_2O_5 hydrolysis reaction in bulk water.

We would like to point out that our calculations explain the conflicting conclusions reached by Galib-Limmer's study (*Science* 371, 921-925 (2021)) and their collaborative research. (*Nat. Commun.* 13, 1266 (2022)) In addition, we provide a quantitative understanding of the uptake mechanism underlying the reactive of N_2O_5 into aqueous

aerosols. More importantly, the strategy and framework developed in this study could have extensions to microdroplets with high chemical complexity.

9) At present, this referee is left guessing as to what is different. The above issues need to be addressed in a thoughtful manner. Let me reiterate that I understand that this is not a referendum on the Galib-Limmer study--Nevertheless, there needs to be something to tell the reader what is different and what can be learned that is different. Just stating that the results are different is not sufficient. Reactions at interfaces are important. Seeking consistency between the two studies and telling the community in a constructive way the potential differences (appropriately caveated) would be useful to provide a roadmap for others to continue and understand this important system.

Author response: We thank the referee for the comment. The reactive uptake of N_2O_5 by aqueous aerosol has long been of particular interest. Galib and Limmer first used neural network potentials to theoretically investigate the reactive uptake of N_2O_5 into an aqueous aerosol. They estimated a reactive uptake coefficient (γ) of 0.6 based on the widely used resistor model, which is an order of magnitude higher than the experimentally derived coefficients (ranging from 0.04 to 0.06). To resolve the inconsistency, they assumed that the evaporation was barrierless and proposed that the uptake was dominated by interfacial processes. However, later MD simulations using MB-nrg potentials challenged the conclusions of the Galib-Limmer study. Note that Galib and Limmer are also authors of the paper. They found that interfacial reaction “*accounts for at most 20%*”, and “*most of the hydrolysis is predicted to take place in bulk water*”. They suggested that “***The disagreement with respect to the neural network model could likely be a failure of the density functional used in the training data***”. This inspired us to use higher quality functionals to investigate the reactive uptake of N_2O_5 by atmospheric aerosol.

Our calculations using high quality functionals give an explanation for the conflicting conclusions reached by Galib-Limmer's study and their collaborative research. Note that the uptake coefficient corresponding to the hydrolysis reaction rates estimated using the widely used resistor model are in good agreement with the experimental data. In addition, we provide a quantitative understanding of the uptake mechanism underlying the reactive uptake of N_2O_5 into aqueous aerosols. More importantly, the strategy and framework developed in this study could be used to study other systems with high quality functionals.

Reviewer #3

N_2O_5 is one of the key reservoir species in both stratospheric and tropospheric NO_x cycles. Its formation and removal rates thus affect both ozone levels, climate and air quality. In this manuscript, Y. G. Fang and co-workers use computational methods to assess the molecular-level details of the main sink reaction of N_2O_5 : uptake into the atmospheric aqueous phase (aqueous aerosols or cloud droplets), and subsequent formation of HNO_3 (and possibly other co-products).

The study reports two main findings: 1) in pure water droplets/aerosol, hydrolysis happens predominantly in the bulk phase rather than at the air-water interface as previously claimed; 2) in the presence of (some unspecified concentration of) ammonia, ammonolysis out-competes hydrolysis. While there are some issues/problems with the employed methods and the data analysis as discussed below, the overall methodology is probably sufficient at least for qualitative work (e.g. a comparison of the three channels discussed above), and the computational results are mostly reproducible with the given details (again, see below for some minor caveats).

The work will thus certainly be of interest at least to atmospheric chemists working with molecular-level modelling. Assessing the broader significance of the work, for example to larger-scale modelling work concerning air quality, stratospheric ozone, or climate, is difficult to do based on the data presented so far. While the information that N_2O_5 hydrolysis happens inside rather than at the surface of droplets is interesting, it is ultimately not particularly important - the net result is still that N_2O_5 is removed by hydrolysis. (This is reminiscent of the question of SO_3 hydrolysis, where several studies have been published demonstrating that various molecules can catalyse the hydrolysis reaction. However, their impact and relevance is limited, as hydrolysis of SO_3 to H_2SO_4 happens on sub-second timescales in any case, in almost any atmospheric conditions.) The second main result is potentially of greater relevance and impact, as it implies that the N_2O_5 removal rate might depend on the NH_3 concentration, and/or possibly the aerosol or droplet composition and pH. (Please see also comment 6 below on quantifying the dependence of the removal rate on the NH_3 concentration). However, even the “slower” quoted bulk hydrolysis rate of $2.3\text{E}-3$ nanoseconds is quite fast - it implies that the lifetime of N_2O_5 in the bulk aqueous phase is less than a microsecond. Does it really matter if N_2O_5 is removed by ammonolysis in less than a nanosecond, or hydrolysis in less than a microsecond, if the net effect is in any case practically instantaneous N_2O_5 removal?

To my thinking, the answer may depend on the fate of the nitramide (NH_2NO_2) product formed in the ammonolysis pathway - is this molecule expected to live long enough to affect atmospheric chemistry (or to form more long-lived products other than HNO_3)? If yes, how? A quick literature search reveals relatively few studies on nitramide oxidation, or even nitramide reactions overall, and none about actual atmospheric oxidation. Tantalisingly, some of those (see e.g. <https://pubs.acs.org/doi/10.1021/acs.est.6b04842> for an environmental chemistry study, albeit on a much larger compound) suggest N_2O as a possible product. Even limited production of N_2O from N_2O_5 in the atmospheric aqueous phase would certainly be “huge if true”. I’m certainly interested in hearing the authors’ reasoning (and possible speculations) here.

Author response: We appreciate the reviewer for the thorough and insightful comments on our manuscript. Over the past few decades, the reactive uptake of N_2O_5 on aerosols has been widely considered one of the most influential processes in heterogeneous chemistry. In this study, in addition to exploring the reaction kinetics of the hydrolysis and ammonolysis of N_2O_5 at the air-water interface and in bulk water, we present a practical framework for studying chemical reactions at the air-water interface, a key reaction site in the atmosphere, which is equally important.

In addition, we would like to point out that the quantitative picture of the uptake mechanisms has long been of particular interest. The reactive uptake of N_2O_5 into an aqueous aerosol was first investigated by Galib and Limmer using neural network potentials. They estimated a reactive uptake coefficient (γ) of 0.6 based on the widely used resistor model, (*Science* 371, 921-925 (2021)) which is an order of magnitude higher than experimentally derived coefficients (ranging from 0.04 to 0.06). To resolve the inconsistency, they assumed that the evaporation was barrierless and proposed that the uptake was dominated by interfacial processes. However, later MD simulations using MB-nrg potentials challenged the conclusions of the Galib-Limmer study. (*Nat. Commun.* 13, 1266 (2022)) Note that Galib and Limmer are also authors of the paper. Our calculations using high quality functionals provide an explanation for this conflicting conclusion.

Finally, we have discussed the effect of NH_3 concentration on the N_2O_5 ammonolysis and its impact on N_2O generation in the revised manuscript. Relevant papers are also cited in the revised manuscript.

Reviewer #3 Specific comments:

1) The basis sets used in the “benchmark” geometry optimisations and energy calculations are rather modest for benchmarking purposes: 6-31+G** only has a single set of polarisation functions, while cc-pVTZ lacks any diffuse functions. Are the authors sure this is good enough for “benchmark quality”?

Author response: We thank the referee for the comment. Recent studies indicated that “for many organic molecules and their transition states, high-level revDSD-PBEP86-D4 and DLPNO-CCSD(T)/(aug-)cc-pVTZ single-point energies usually vary by less than 2 kcal mol⁻¹ on density functional theory geometries optimized using basis sets ranging from 6-31+G(d) to aug-pcseg-2 and aug-cc-pVTZ” (*J. Chem. Theory Comput.* 19, 5036–5046 (2023)). Therefore, it is reasonable to use the 6-31+G** basis set for geometry optimization. As suggested by the referee, we employed the diffuse basis set aug-cc-pVTZ for single-point calculations at the CCSD(T) level. Fig. R9 displays the reaction barriers for N₂O₅ hydrolysis and ammonolysis in the gas phase calculated at the CCSD(T)/aug-cc-pVTZ//PBE0/6-31+G** level of theory. It can be seen that the reaction barriers calculated at the CCSD(T)/aug-cc-pVTZ//PBE0/6-31+G** level of theory are 0.9 - 2.5 kcal/mol lower than those calculated at the CCSD(T)/cc-pVTZ//PBE0/6-31+G** level of theory. In other words, the difference between the reaction barriers calculated at the PBE0/6-31+G** level and those calculated at the CCSD(T)/aug-cc-pVTZ//PBE0/6-31+G** level relative to CCSD(T)/cc-pVTZ//PBE0/6-31+G** is smaller. In the revised manuscript, we have replaced the results obtained with CCSD(T)/cc-pVTZ method with those obtained from the CCSD(T)/aug-cc-pVTZ method.

Fig. R9. Reaction barriers for N₂O₅ hydrolysis and ammonolysis in the gas phase calculated at the CCSD(T)/aug-cc-pVTZ or CCSD(T)/cc-pVTZ level of theory.

2) The three sets of numbers given in the first paragraph of the “Results and Discussion” section don’t appear to match each other. For example, for the 1W-1 reaction, the energy barrier computed at “high levels of theory” is given as 14.3 to 16.8 kcal/mol. The rev-PBE barrier is then given as 8.6 kcal/mol. Finally, it is stated that this is “13.0 kcal/mol lower” than at the “benchmark” level. However, 8.6 is not 13 lower than 14.3-16.8. The same applies to all the three other reactions - the quoted difference does not match the two previous numbers. Please clarify this.

Author response: We thank the referee for the comment and apologize for the lack of clarity. In the original text, we categorized density functional methods BP86, BLYP, PBE, and revPBE as “low-level” methods (depicted by blue bars), while B3LYP, PBE0, M06, and B2PLYP are considered as “high-level” methods (represented by red bars). The CCSD(T) results are represented by dashed lines in Fig. 1. For the 1W-1 reaction, the calculated barrier obtained using the revPBE functional is 8.6 kcal/mol, which underestimates the CCSD(T) result by about 13.0 kcal/mol. Since CCSD(T) serves as the gold standard, we compare the results obtained at different functional levels with CCSD(T) instead of PBE0 etc. In order to enhance the clarity, we have made corresponding changes in the revised manuscript.

3) I don’t understand how Figure 1b and d can be interpreted to suggest that “revPBE performed the best”. Especially for the ammonolysis reaction, revPBE performs very poorly, much worse than B3LYP, PBE0 or M02X, all of which are methods with roughly similar computational expense. (B2PLYP is much more expensive.) Is the argument here that revPBE is somehow a “low-level” method, while e.g. B3LYP is “high level”? Are the red columns in the figure supposed to be “low level”, and the blue columns “high”? This division seems arbitrary and even incorrect, and even if it were true, revPBE is not really much better than the other “red” methods (which are all atrociously bad).

Author response: We thank the referee for the comment and apologize for the lack of clarity. We agree with the reviewer that the revPBE functional has a poor overall performance in describing hydrolysis and ammonolysis reactions compared to functionals such as PBE0 and B3LYP. However, the mechanism of uptake has been explored using a neural network potential, which was fitted based on the revPBE functional. ((*Science* 371, 921-925 (2021))) As a revised version of the PBE functional, compared to other generalized gradient approximation (GGA) functionals such as PBE, BLYP, and BP86, the revPBE functional exhibits only a slight improvement in describing the energy barriers of hydrolysis and ammonolysis reactions. Hence, it is still necessary to choose the hybrid functional.

Furthermore, as pointed out by the reviewer, the descriptions of "high-level" and "low-level" are unclear and ambiguous. In the revised manuscript, we have replaced "low-level" with the terminology "generalized gradient approximation functionals (specifically BP86, BLYP, PBE, revPBE)" and "high-level" with "hybrid functionals (including B3LYP, PBE0, M06, B2PLYP, where B2PLYP represents a double-hybrid functional)". We have made corresponding changes in the revised manuscript.

4) Do any of the entries in Figures 1b and d correspond to the methods actually used in the QM/MM simulations (PBE-D3/DZVP-MOLOPT-SR and PBE0-D3/DZVP, according to the methods - section)? If yes, please label it accordingly - if no, please redo the calculations and include this method in the figure!

Author response: We thank the referee for the comment. The MOLOPT basis set is a unique Gaussian-type basis set supported by CP2K. Note that it performs particularly well for a large number of systems and also targets a wide range of chemical environments (*J. Chem. Phys.* 127, 114105 (2007); *J. Chem. Phys.* 152, 194103 (2020)). Since the transition state optimization algorithm in CP2K is not as efficient as the one in Gaussian software, we used Gaussian software to explore the production of HNO₃ via N₂O₅ hydrolysis and ammonolysis in the gas phase. DZVP-MOLOPT-SR and 6-31+G** are polarized double-zeta basis set which yield similar results. As suggested by the referee, we performed calculations using the DZVP-MOLOPT-SR basis set and obtained almost the same results as using the 6-31+G** basis set (Figure. R10). We have modified the main text accordingly.

Fig. R10. Calculated DFT energy barriers ΔE_b at the PBE0/DZVP-MOLOPT-SR//PBE0/6-31+G* level for the (a) hydrolysis and (b) ammonolysis reaction of N₂O₅ for all the functionals considered.

5) Please explain from which data the transition-state theory rates are computed. I assume it is the PBE0-D3/DZVP data in the “high-level” QM/MM simulations? Given the rather large differences between PBE and CCSD(T) in Figures 1b and 1d, how reliable should the absolute rates be considered? Should they perhaps be reduced by some correction factor proportional to $\exp(-dE/RT)$, where dE is the difference in barrier heights between the methods? (Note that the answer to the previous question may render this question either more or less relevant.)

Author response: We thank the referee for the comment. The transition-state theory rates were computed using the PBE0-D3/DZVP-MOLOPT-SR data obtained from QM/MM simulations. The energy barriers (ΔE_b) for the gas phase reactions calculated at PBE0/DZVP-MOLOPT-SR level are lower than those calculated at the CCSD(T)/aug-cc-PVTZ//PBE0/DZVP-MOLOPT-SR level. However, the performance of GGA functionals is very poor, while hybrid functionals perform much better than GGA functionals. Considering that it is impossible to use the CCSD(T) functional in QM/MM MD simulations because of the high computational cost, we used the PBE0 functional.

It is noteworthy that the difference in ΔE_b between the gas phase reactions calculated at PBE0/DZVP-MOLOPT-SR level and those calculated at the CCSD(T)/aug-cc-PVTZ//PBE0/DZVP-MOLOPT-SR level decreases with the increase of water molecules. Specifically, for reactions in which an H group of H_2O is attached to the terminal oxygen atom of O_2NONO_2 , i.e., pathways 1W-1 and 2W-1, as the number of water molecules increases from one to two, the difference in ΔE_b between the gas phase reactions calculated at PBE0/DZVP-MOLOPT-SR level and those calculated at the CCSD(T)/aug-cc-PVTZ//PBE0/DZVP-MOLOPT-SR level decreases from 5.3 to 4.3 kcal/mol. It is expected that the hydrolysis reaction does not require any correction factor due to the large number of water molecules present in liquid water. In fact, the reactive uptake coefficient predicted using the uncorrected transition-state theory rates agrees well with experimental results.

6) The hydrolysis rate is expressed in terms of a unimolecular rate coefficient (in units of 1/ns). While the reaction between N_2O_5 and H_2O is of course bimolecular, in an aqueous-phase (bulk or interface) context this is appropriate: the concentration of water is after all close to constant (even in relatively concentrated solutions), and it makes sense to implicitly include it in the rate coefficient. However, for ammonolysis this is not the case - the rate at which N_2O_5

is ammonolysed will certainly depend on the ammonia concentration, so giving one single pseudo-unimolecular number is meaningless. Please provide instead a figure or table of the pseudo-unimolecular ammonolysis rate as a function of the liquid-phase ammonia concentration, or even the gas-phase ammonia concentration. The latter actually becomes an interesting exercise, as the $[\text{NH}_3]_{\text{aq}}$ depends not only on p_{NH_3} but also on pH - unless also NH_4^+ ions can ammonolyse N_2O_5 ? As a side note, the results would imply a possible pH - dependence of the N_2O_5 uptake coefficient, which would be intriguing, albeit with two caveats: 1) also the hydrolysis lifetime is so short that this dependence may not matter; and 2) just like in the case of SO_3 mentioned above, it may well turn out that aqueous-phase N_2O_5 destruction is catalysed by many different species, including both acidic and basic molecules. (The latter argument is certainly beyond the scope of the present manuscript; I'm just raising the issue as something the authors might want to look at later).

Author response: We thank the reviewer for the comment. Our calculations show that the interfacial ammonolysis rate is very fast, i.e., $\sim 151 \text{ ns}^{-1}$, which indicates that the interfacial ammonolysis of N_2O_5 is barrierless. Under such conditions, the rate constants can be evaluated using collision frequency model. Assuming concentration of the ammonia is $n \text{ mol/L}$, the ammonia concentration at the air-water interface is given by

$$c = n \exp(\beta\Delta F_b) \quad (1)$$

where $\beta\Delta F_b = 1.54$ is the barrier to move from the bulk liquid to the interface (*Proc. Natl. Acad. Sci.* 115, 6147-6152 (2018)). Then, an expression for the collision frequency of each N_2O_5 molecule is obtained:

$$k = 1.54nN_A\pi(r_{\text{NH}_3} + r_{\text{N}_2\text{O}_5})^2 \sqrt{v_{\text{N}_2\text{O}_5}^2 + v_{\text{NH}_3}^2} \quad (2)$$

where N_A is the Avogadro constant, $r_{\text{NH}_3} = 0.20 \text{ nm}$ is the radius of NH_3 , $r_{\text{N}_2\text{O}_5} = 0.35 \text{ nm}$ is the radius of N_2O_5 , $v_{\text{N}_2\text{O}_5} = 15.8 \text{ nm/ns}$ is the average velocity of N_2O_5 , and $v_{\text{NH}_3} = 15.9 \text{ nm/ns}$ is the average velocity of NH_3 . Then $k = 2.02 \times 10^{10}n \text{ s}^{-1}$. In order to have a clear picture of ammonolysis rates as a function of NH_3 concentration (n), we plotted k against n , as shown in Fig. R11. The plot clearly shows that the ammonolysis rate increases monotonically with an increase in n . Moreover, it is evident that ammonolysis competes with hydrolysis at NH_3 concentrations above $1.9 \times 10^{-4} \text{ mol/L}$. Note that Recent satellite measurements and integrated cross-scale modeling show that ammonia tends to accumulate on the surface of cloud droplets (*Proc. Natl. Acad. Sci. U. S. A.* 115, 6147-6152 (2018)). Thus, interfacial ammonolysis of N_2O_5 may be important, especially in highly polluted regions.

Fig. R11. Reaction rate of ammonolysis as a function of NH₃ concentration. Upper and lower hydrolysis rate of N₂O₅ in liquid water are represented by dashed line.

7) The authors are apparently comparing their pure-water uptake coefficients to measurements done for ammonium sulfate and bisulfate particles (discussion around Figure 6). However, their central argument in this study is that ammonia affects the uptake process! Does this not invalidate the comparison?

Author response: We thank the reviewer for the comment. Our calculations predict that the reactive uptake coefficient (γ) on pure water ranges from 0.01 to 0.07, which is in good agreement with the experimental results, i.e., γ of N₂O₅ in pure water is in the range of 0.04 and 0.06.¹ Unlike the reaction of N₂O₅ on pure water, the heterogeneous reaction of N₂O₅ on aqueous NH₄HSO₄ and (NH₄)₂SO₄ particles was widely reported.² In these studies, measurements showed that γ (N₂O₅) on aqueous NH₄HSO₄ and (NH₄)₂SO₄ particles ranged from 0.01 to 0.1. Interestingly, γ increases with increasing RH, although the reliability of this trend at high RH is unclear,¹⁻² which maybe due to the fact that the concentration of NH₃ increases with increasing RH as NH₄⁺ is hydrolysed. We have made corresponding changes in the revised manuscript.

Reference:

1. Kane SM, et al. Heterogeneous uptake of gaseous N₂O₅ by (NH₄)₂SO₄, NH₄HSO₄, and H₂SO₄ aerosols. *J. Phys. Chem. A* 105, 6465-6470 (2001).
2. Davis JM, et al. Parameterization of N₂O₅ reaction probabilities on the surface of particles containing ammonium, sulfate, and nitrate. *Atmos. Chem. Phys.* 8, 5295-5311 (2008).

8) Please explain a bit more what the transmission coefficient $k(t)$ in 1 accounts for. Also, what

does the “(t)” notation here mean - how is the coefficient time-dependent?

Author response: We thank the reviewer for the comment. Transition state theory (TST) assumes that the trajectory moves through the transition state undeterred (i.e., the activated trajectory will not recross the transition state). In fact, the active trajectory can recross the transition state. The corrections to the TST estimate of rate constants are conveniently expressed in terms of the transmission coefficient, k . It is defined by

$$k_{\text{true}} = k^* k_{\text{TST}} \quad (1)$$

and roughly speaking, k , is the fraction of successful trajectories. For those trajectories that are in the transition state at $t = 0$, typical transient dynamics away from it and towards a stable situation will occur in a relatively rapid time $t \sim \tau_{\text{mol}}$. This idea leads one to the reactive flux correlation function defined as follows (*J. Stat. Phys.* 42, 49-67, (1986)):

$$k(t) = \langle v(0) \delta[q(0) - q^*] H_B[q(t)] \rangle \quad (2)$$

where, $q(t)$ is the reaction coordinate at time t ; $v(t) = \dot{q}(t)$ is the velocity of that coordinate; $H_B[q(t)]$ is the characteristic function for stable state B, i.e., it is 1 for $q(t) > q^*$ and it is zero otherwise. The angle brackets indicate the equilibrium ensemble average over the initial conditions of all degrees of freedom.

REVIEWERS' COMMENTS

Reviewer #1 (Remarks to the Author):

The authors did a careful and extensive work to address the reviewers' criticisms and improved substantially the quality of the manuscript. In particular, they provide convincing motivation for their technical choices and enable reproducibility of their data. They also improved the figures and the discussion of their results.

I recommend that this paper is published in Nature Communications.

Reviewer #4 (Remarks to the Author):

In response to comments by myself and two other reviewers, the authors have performed additional simulations, and considerably modified the manuscript. They have also provided extensive explanations to our various questions in their rebuttal letter. I thank them for their thoroughness in this. The manuscript is now much improved. While I still feel it is slightly exaggerated to call the results "quantitative" given the huge discrepancies in barrier heights between the benchmark coupled-cluster simulations and the methods actually applied here, I am convinced that the work here is state-of-the-art, and have no further comments or requests concerning the methodological aspects. (In any case, the other reviewers seem to have more expertise in the intricacies of e.g. the metadynamics approaches.)

My one remaining question concerns the NH_3 -concentration dependence, which the authors have now helpfully provided. It seems the pseudo-unimolecular rate reported e.g. in the abstract corresponds to a system where the NH_3 is already in contact with an N_2O_5 , so converting this into a bimolecular rate (or into a $[\text{NH}_3]$ - dependent unimolecular rate) is fairly straightforward, and involves mainly collision kinetics. So far so good. However, the break-even point where the ammonolysis rate is equal to the hydrolysis rate is reported to correspond to 1.9×10^{-4} mol/L. This is a very high ammonia concentration. The Henry's law constant of NH_3 (from NIST, at 298 K - other sources give fairly similar values as well) is roughly 60 mol/L atm (with a modest temperature dependence, e.g. at 273 K I get a value of about 220). So a liquid-phase concentration of 1.9×10^{-4} mol/L would require a gas-phase

concentration of about 3×10^{-6} atm - or about 3000 ppb. While such concentrations are sometimes encountered close to intense pollution sources (e.g. cattle farms, <https://www.tandfonline.com/doi/abs/10.3155/1047-3289.60.2.210>), even polluted urban air usually doesn't contain more than a few tens of ppbs of ammonia, and the typical background continental mixing ratio is in the single-digit ppbs (see e.g. <https://catalog.data.gov/dataset/atmospheric-composition-ammonia-volume-mixing-ratio-l3-airnac3mnh3-v3-from-airnac-amsu-on-na>). So taking the results at face value, ammonolysis would typically not contribute more than some fractions of a percent of the total N₂O₅ decomposition. The authors cite sources showing possible surface enrichment of NH₃, so it might be that their model underestimates the interfacial NH₃ concentration corresponding to a certain bulk value. (Especially at lower temperatures - with higher NH₃ solubilities - the break-even threshold is after all not TOO far off the actual expected [NH₃] range.) However, this should then be explicitly discussed, and the implications of the computed concentration dependence should be admitted already in the abstract (rather than using the pseudo-unimolecular rates to imply that ammonolysis is dominant, which does not seem to be the case).

Based on the comments of Reviewer #4, we have added a relevant discussion on N_2O_5 ammonolysis in the revised manuscript.

Reviewer #1 (Remarks to the Author):

The authors did a careful and extensive work to address the reviewers' criticisms and improved substantially the quality of the manuscript. In particular, they provide convincing motivation for their technical choices and enable reproducibility of their data. They also improved the figures and the discussion of their results.

I recommend that this paper is published in Nature Communications.

Author response: We appreciate the reviewer for suggesting this paper for publication in Nature Communications.

Reviewer #4 (Remarks to the Author):

In response to comments by myself and two other reviewers, the authors have performed additional simulations, and considerably modified the manuscript. They have also provided extensive explanations to our various questions in their rebuttal letter. I thank them for their thoroughness in this. The manuscript is now much improved. While I still feel it is slightly exaggerated to call the results "quantitative" given the huge discrepancies in barrier heights between the benchmark coupled-cluster simulations and the methods actually applied here, I am convinced that the work here is state-of-the-art, and have no further comments or requests concerning the methodological aspects. (In any case, the other reviewers seem to have more expertise in the intricacies of e.g. the metadynamics approaches.)

Author response: We appreciate the reviewer for the positive evaluation of our work.

My one remaining question concerns the NH_3 -concentration dependence, which the authors have now helpfully provided. It seems the pseudo-unimolecular rate reported e.g. in the abstract corresponds to a system where the NH_3 is already in contact with an N_2O_5 , so converting this into a bimolecular rate (or into a $[\text{NH}_3]$ -dependent unimolecular rate) is fairly straightforward, and involves mainly collision kinetics. So far so good. However, the break-even point where the ammonolysis rate is equal to the hydrolysis rate is reported to correspond to 1.9×10^{-4} mol/L. This is a very high ammonia concentration. The Henry's law constant of NH_3 (from NIST, at 298 K - other sources give fairly similar values as well) is roughly 60 mol/L atm (with a modest temperature dependence, e.g. at 273 K I get a value of about 220). So a liquid-phase concentration of 1.9×10^{-4} mol/L would require a gas-phase concentration of about 3×10^{-6} atm - or about 3000 ppb. While such concentrations are sometimes encountered close to intense pollution sources (e.g. cattle farms, <https://www.tandfonline.com/doi/abs/10.3155/1047-3289.60.2.210>), even polluted urban air usually doesn't contain more than a few tens of ppbs of ammonia, and the typical background continental mixing ratio is in the single-digit ppbs (see e.g. <https://catalog.data.gov/dataset/atmospheric-composition-ammonia-volume-mixing-ratio-l3-airsac3mnh3-v3-from-airs-amsu-on-na>). So taking the results at face value,

ammonolysis would typically not contribute more than some fractions of a percent of the total N_2O_5 decomposition. The authors cite sources showing possible surface enrichment of NH_3 , so it might be that their model underestimates the interfacial NH_3 concentration corresponding to a certain bulk value. (Especially at lower temperatures - with higher NH_3 solubilities - the break-even threshold is after all not TOO far off the actual expected $[\text{NH}_3]$ range.) However, this should then be explicitly discussed, and the implications of the computed concentration dependence should be admitted already in the abstract (rather than using the pseudo-unimolecular rates to imply that ammonolysis is dominant, which does not seem to be the case).

Author response: We thank the reviewer for the comment. Based on the comments of the reviewer, we have added a relevant discussion on N_2O_5 ammonolysis to the revised manuscript. Additionally, we have included the effect of ammonia concentration on the rate of N_2O_5 ammonolysis in the abstract.